# Niemann-Pick type C proteins promote microautophagy by expanding raft-like membrane domains in the yeast vacuole

**Takuma Tsuji, Megumi Fujimoto, Tsuyako Tatematsu, Jinglei Cheng, Minami Orii, Sho Takatori[†], Toyoshi Fujimoto\***

Department of Molecular Cell Biology and Anatomy, Nagoya University Graduate School of Medicine, Nagoya, Japan

**Abstract** Niemann-Pick type C is a storage disease caused by dysfunction of NPC proteins, which transport cholesterol from the lumen of lysosomes to the limiting membrane of that compartment. Using freeze fracture electron microscopy, we show here that the yeast NPC orthologs, Ncr1p and Npc2p, are essential for formation and expansion of raft-like domains in the vacuolar (lysosome) membrane, both in stationary phase and in acute nitrogen starvation. Moreover, the expanded raft-like domains engulf lipid droplets by a microautophagic mechanism. We also found that the multivesicular body pathway plays a crucial role in microautophagy in acute nitrogen starvation by delivering sterol to the vacuole. These data show that NPC proteins promote microautophagy in stationary phase and under nitrogen starvation conditions, likely by increasing sterol in the limiting membrane of the vacuole.

**\*For correspondence:** tfujimot@ med.nagoya-u.ac.jp

**Present address:** [†]Laboratory of Neuropathology and Neuroscience, Graduate School of Pharmaceutical Sciences, The University of Tokyo, Tokyo, Japan

**Competing interests:** The authors declare that no competing interests exist.

## Introduction

The Niemann-Pick type C (NPC) proteins NPC1 and NPC2 bind with cholesterol derived from endo-cytosed lipoproteins and transport it to the lysosomal membrane (*Kwon et al., 2009*; *Wang et al., 2010*). Genetic defects in NPC1 and NPC2 cause a neurodegenerative disorder characterized by the accumulation of cholesterol and other lipids in lysosomes, demonstrating the physiological importance of these proteins (*Rosenbaum and Maxfield, 2011*).

The molecular mechanism by which NPC proteins transport cholesterol has been studied extensively: NPC2, a soluble protein, binds cholesterol and transfers it to NPC1, a multipass transmembrane protein, which then inserts cholesterol into the lysosomal membrane (*Kwon et al., 2009*; *Wang et al., 2010*). In parallel with this 'hand-off' mechanism, NPC2 is also thought to transport cholesterol directly to the membrane, bypassing NPC1 (*Kennedy et al., 2012*; *Xu et al., 2008*). Cholesterol that is inserted into the lysosomal membrane is then delivered to other organelles, including the endoplasmic reticulum and the plasma membrane, by mechanisms that are yet to be defined in detail (*Ikonen, 2008*).

At one point during this transport process, cholesterol must be incorporated into the lysosomal membrane. Considering the low cholesterol content of the lysosomal membrane under ordinary circumstances (*Kolter and Sandhoff, 2005*) and the known effect of cholesterol on various membrane properties (*Bloom et al., 1991*; *Lingwood and Simons, 2010*), the insertion of cholesterol by NPC proteins is likely to influence the lysosomal membrane significantly, but to the best of our knowledge, relatively little attention has been paid to this aspect.

In the present study, we aimed to investigate this matter through experiments with budding yeast *Saccharomyces cerevisiae*. The use of yeast is legitimate because the functional orthologs of NPC proteins in budding yeast, namely, Ncr1p and Npc2p, can substitute for NPC1 and NPC2,

**eLife digest** Niemann-Pick disease type C is a human disease that is characterized by severe neurological symptoms. The brains of children with this disease develop more slowly and adult patients have difficulty walking, coordinating movements, speaking clearly and they develop dementia. The disease is caused by mutations in two proteins called NPC1 and NPC2, which are normally needed to move lipid molecules, especially cholesterol, between different compartments within a cell.

Lysosomes are compartments within human cells that act as recycling points for many nutrients, including lipid molecules. The membranes surrounding lysosomes can bend to form pouches that can engulf the materials to be recycled. However, it was not clear exactly how this process, also known as microautophagy, happens and whether the NPC1 and NPC2 are involved.

Here Tsuji et al. used a method called freeze fracture to study microautophagy in yeast under an electron microscope. For the experiments, the yeast cells were exposed to conditions that prevented them from dividing to mimic human nerve cells in the brain. The experiments show that NPC1 and NPC2 play important roles in creating and enlarging areas in the lysosome's membranes that are rich in a lipid molecule called ergosterol, which yeast uses in the same way as animals use cholesterol. These areas, also known as rafts, promoted microautophagy of lipid storage compartments in the yeast cells, ensuring the healthy recycling of nutrients. Furthermore, when the normal version of NPC2 was replaced with a mutated form of NPC2 that could not bind to ergosterol, the yeast cells were less able to form ergosterol-rich rafts.

These findings indicate that the symptoms of Niemann-Pick type C may be due, at least in part, to defects in the formation of cholesterol-rich lipid rafts in the membranes of lysosomes. Future experiments may investigate whether this microautophagy process, which depends on the NPC proteins in yeast, is exactly the same in human cells.

respectively, in mammalian cells (*Berger et al., 2005*; *Malathi et al., 2004*). We examined yeast in two different conditions: one is the stationary phase, in which the cell cycle is arrested because of the gradual exhaustion of nutrients, and in which the formation of raft-like domains and lipophagy occur in the vacuole (*Toulmay and Prinz, 2013*; *Wang et al., 2014*); the other is acute nitrogen starvation, which causes macroautophagy as well as microautophagy of lipid droplets (LDs) and the nucleus (*Roberts et al., 2003*; *van Zutphen et al., 2014*).

We discovered that NPC proteins, especially Npc2p, play a critical role in the formation and expansion of the raft-like domain in the vacuole, and that the domain is engaged in the binding and engulfment of LDs, both in stationary phase and in acute nitrogen starvation. This result indicates that a major function of NPC proteins is to expand the sterol-rich raft-like membrane domain in the vacuole to execute microautophagy.

## Results

### Lipophagy in stationary phase occurs through the expansion of raft-like vacuolar membrane domains

In stationary phase yeast, raft-like domain formation and lipophagy have been hypothesized to proceed in a feed-forward manner (*Wang et al., 2014*), but the exact mechanism has not been clear. To determine how the raft-like domain is involved in lipophagy, we observed the entire process by means of freeze-fracture electron microscopy (EM).

The vacuolar membrane in stationary phase showed geometrical patterns (*Moeller and Thomson, 1979*) (*Figure 1A*). We confirmed that domains that are deficient in the intramembrane particles (IMPs), which represent transmembrane proteins, correspond to the raft-like domain by labeling the non-raft domain marker Vph1p–mRFP in freeze-fracture replicas (*Toulmay and Prinz, 2013*) (*Figure 1B* and *Figure 1—figure supplement 1A*).

To examine how the vacuolar domains are related to lipophagy, we performed freeze-etching after freeze-fracture to make LDs clearly identifiable in EM (*Figure 1—figure supplement 1B*). This

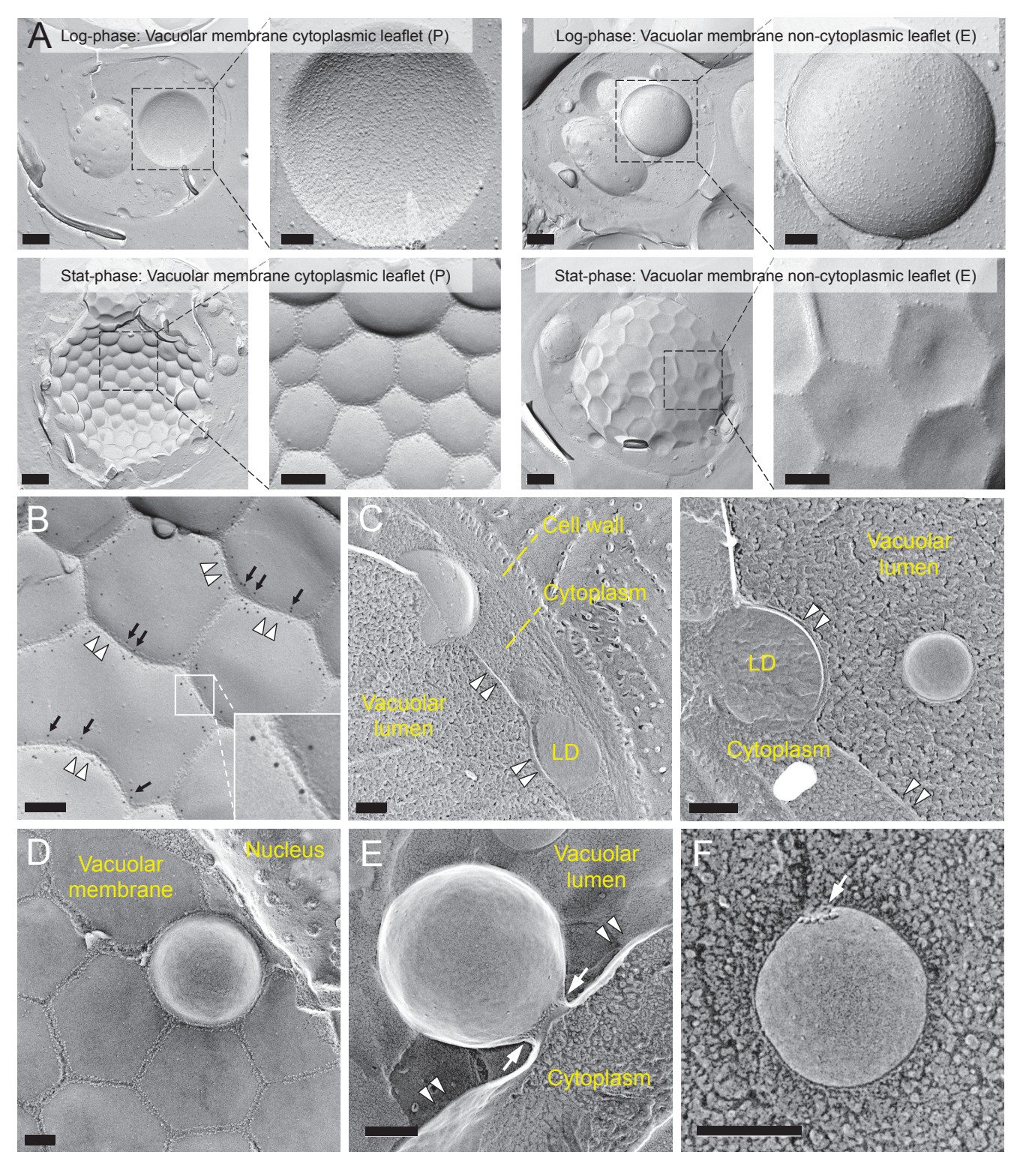

**Figure 1.** Lipophagy in stationary phase occurs through the expansion of raft-like vacuolar membrane domains. (**A**) Freeze-fracture EM of the vacuole. The vacuolar membrane in stationary phase yeast shows geometrical patterns in both the protoplasmic face (P face; cytoplasmic leaflet) and the exoplasmic face (E face: non-cytoplasmic leaflet). Intramembrane particles (IMPs), which represent transmembrane proteins, are largely confined to the edges of polygonal areas. Bars: 0.5 μm (low magnification); 0.2 μm (high magnification). (**B**) Freeze-fracture labeling of Vph1p–mRFP in the stationary phase vacuole. Labels for Vph1p–mRFP (arrows) were excluded from the polygonal IMP-deficient area, but were present in the IMP-rich

*Figure 1 continued on next page*

*Figure 1 continued*

region, indicating that the IMP-deficient area corresponds to the raft-like domain. The IMP-rich area, or the nonraft-like domain, is marked by double arrowheads. Bar: 0.2 μm. (C–F) Freeze-fracture/etching EM of stationary phase vacuole. Bars: 0.2 μm. The black-and-white contrast is reversed to enhance the three-dimensional features. (C) LDs adhered tightly to IMP-deficient raft-like domain, which bulges toward the lumen. The vacuolar membrane is marked by double arrowheads. See *Figure 1—figure supplement 2* for a picture of the whole cell and *Video 1* for a three-dimensional view of the LD-vacuole apposition. (D) Only a limited number of raft-like domains showed bulging. Irrespective of the variable degree of bulging in raft-like domains, nonraft-like domains were confined to the edge. See *Figure 1—figure supplement 3* for additional photos. (E) The raft-like domain formed a balloon-like structure that protrudes into the vacuolar lumen. IMPs were observed as a ring-like structure in the constricted portion (arrows). The vacuolar membrane is marked by double arrowheads. (F) Vesicles in the vacuolar lumen harbored IMP clusters (arrow).

The following figure supplements are available for figure 1:

**Figure supplement 1.** The procedure for quick-freezing and freeze-fracture EM.

**Figure supplement 2.** Freeze-fracture/etching EM of an entire yeast cell in stationary phase.

**Figure supplement 3.** Additional examples of raft-like domain bulging in the vacuole.

---

technique revealed that the invaginating IMP-deficient raft-like domain, which is tightly bound to LDs (*Figure 1C*, *Figure 1—figure supplement 2*, and *Video 1*), expanded and bulged toward the lumen and formed balloon-like structures, whereas the IMP-rich, nonraft-like domain was confined to the edges of bulges and made a ring-like structure at the neck of the raft-like domain balloon (*Figure 1D,E*, and *Figure 1—figure supplement 3*). IMP clusters were also observed in vesicles in the vacuolar lumen (*Figure 1F*), indicating that the balloon-like structures were pinched off at the nonraft-like domain to complete the microautophagic process (hereafter the intravacuolar vesicles formed by microautophagy will be called microautophagic vesicles).

The distribution of phosphatidylinositol 3-phosphate [PtdIns(3)P], as revealed by freeze-fracture replica labeling, was consistent with the microautophagic mechanism. That is, PtdIns(3)P was confined to the cytoplasmic leaflet both in the vacuolar membrane (*Figure 2A* and *Figure 2—figure supplement 1A*) and in the microautophagic vesicles (*Figure 2B,C*, and *Figure 2—figure supplement 1B*). This PtdIns(3)P asymmetry was opposite to that found in autophagic bodies in macroautophagy (hereafter called macroautophagic bodies), in which PtdIns(3)P is far more abundant in the non-cytoplasmic leaflet than in the cytoplasmic leaflet (*Cheng et al., 2014*) (*Figure 2—figure supplement 1C*), and was utilized to distinguish microautophagic vesicles from other structures in subsequent experiments. These results demonstrated that stationary phase lipophagy is a microautophagic process made possible by expansion of the raft-like domain (*Figure 2D*).

## NPC proteins are essential for stationary phase lipophagy

Consistent with the formation of raft-like domains, sterol stained by filipin was observed along the vacuolar membrane in stationary phase, giving a double-ring staining pattern, whereas filipin staining in log-phase yeast was found only in the cell surface (*Figure 3—figure supplement 1A*). Freeze-fracture EM confirmed the presence of filipin-sterol complexes in the vacuolar raft-like domain (*Figure 3—figure supplement 1B*) (*Elias et al.,*

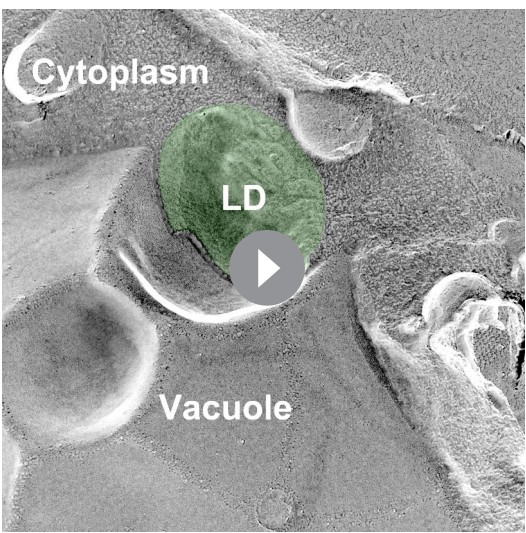

**Video 1.** Tilted images of a freeze-fracture/etching replica of stationary phase yeast. An LD revealing the non-etchable content is adhered to the IMP-deficient domain of the vacuolar membrane.

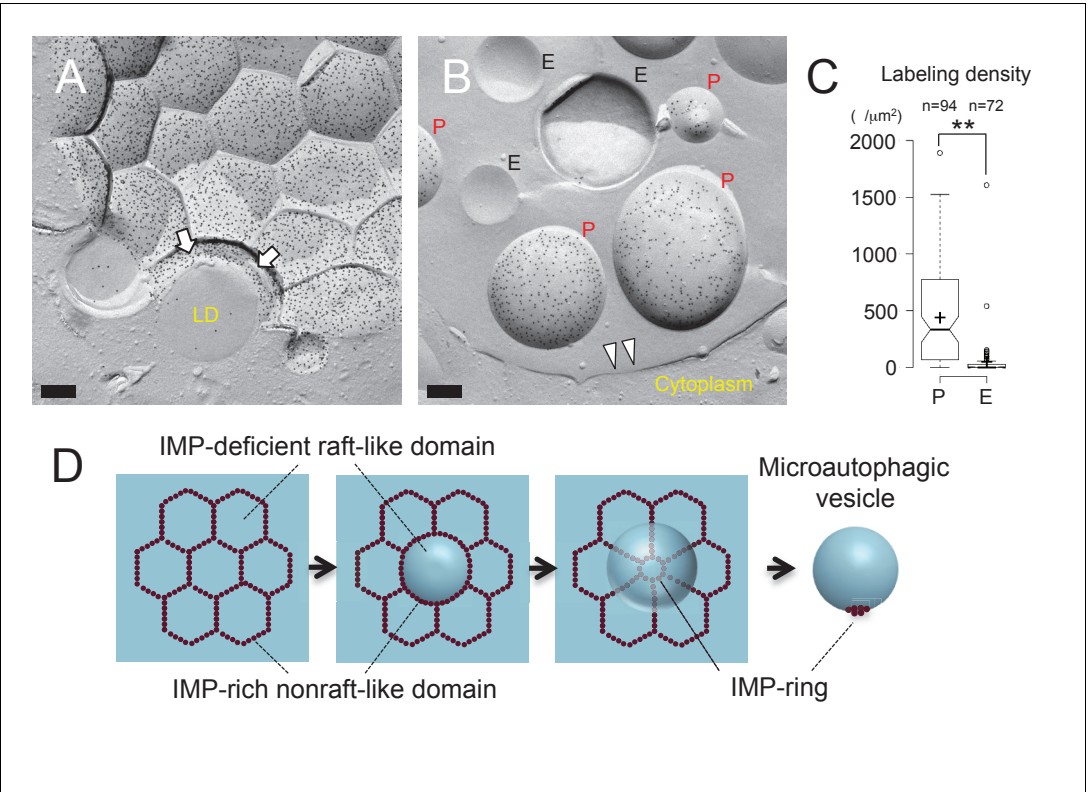

**Figure 2.** Microautophagic vesicles show the same PtdIns(3)P asymmetry as the vacuolar membrane. (**A, B**) Freeze-fracture replica labeling of PtdIns(3)P in stationary phase vacuoles. Colloidal gold particles indicate PtdIns(3)P labeled by recombinant GST-p40phox PX domain. Bars: 0.2 μm. See also *Figure 2—figure supplements 1A and B* for additional photos. (**A**) The vacuolar membrane was labeled for PtdIns(3)P in the cytoplasmic leaflet. Note that the domain engulfing an LD is also labeled (arrows). (**B**) PtdIns(3)P was labeled densely in the convex protoplasmic face (P; the cytoplasmic leaflet), but scarcely in the concave exoplasmic face (E; the luminal leaflet), of microautophagic vesicles. The vacuolar membrane is marked by double arrowheads. (**C**) PtdIns(3)P labeling density in the intravacuolar vesicles. The density was significantly higher in the P face than in the E face. The result of one representative experiment out of three independent experiments. \*\*p<0.01. (**D**) Diagram showing how the vacuolar membrane forms the microautophagic vesicle in stationary phase.

The following figure supplement is available for figure 2:

**Figure supplement 1.** PtdIns(3)P in a stationary phase vacuole.

1979), indicating that the internal filipin staining in stationary phase yeast is derived from the vacuolar membrane at least partially. Although filipin staining shows the relative distribution, but not the absolute amount, of sterols in membranes, the result suggested an increase of sterol in the vacuolar membrane in stationary phase. We hypothesized that this probable increase in sterol in the stationary phase vacuolar membrane may not occur through spontaneous incorporation but may involve NPC proteins; accordingly, we examined $ncr1\triangle$, $npc2\triangle$, and $ncr1\triangle npc2\triangle$ to test this idea.

First, domain formation in the vacuolar membrane was compared by freeze-fracture EM (*Figure 3A*). In wild-type cells (WT), a majority of vacuoles showed domains with an inward curvature (type II). In $ncr1\triangle$, $npc2\triangle$, and $ncr1\triangle npc2\triangle$, the proportion of vacuoles with no domain was greater, whereas the proportion with the type II vacuole was significantly lower (*Figure 3A*).

Second, lipophagy in stationary phase was examined by fluorescence microscopy as described previously (*Wang et al., 2014*). The proportion of cells showing LDs in the vacuole was significantly smaller in NPC-deficient cells than in WT (*Figure 3B*). Here it is worth noting that the number and sizes of LDs were not drastically different between WT and NPC-deficient cells (*Figure 3—figure supplement 2*).

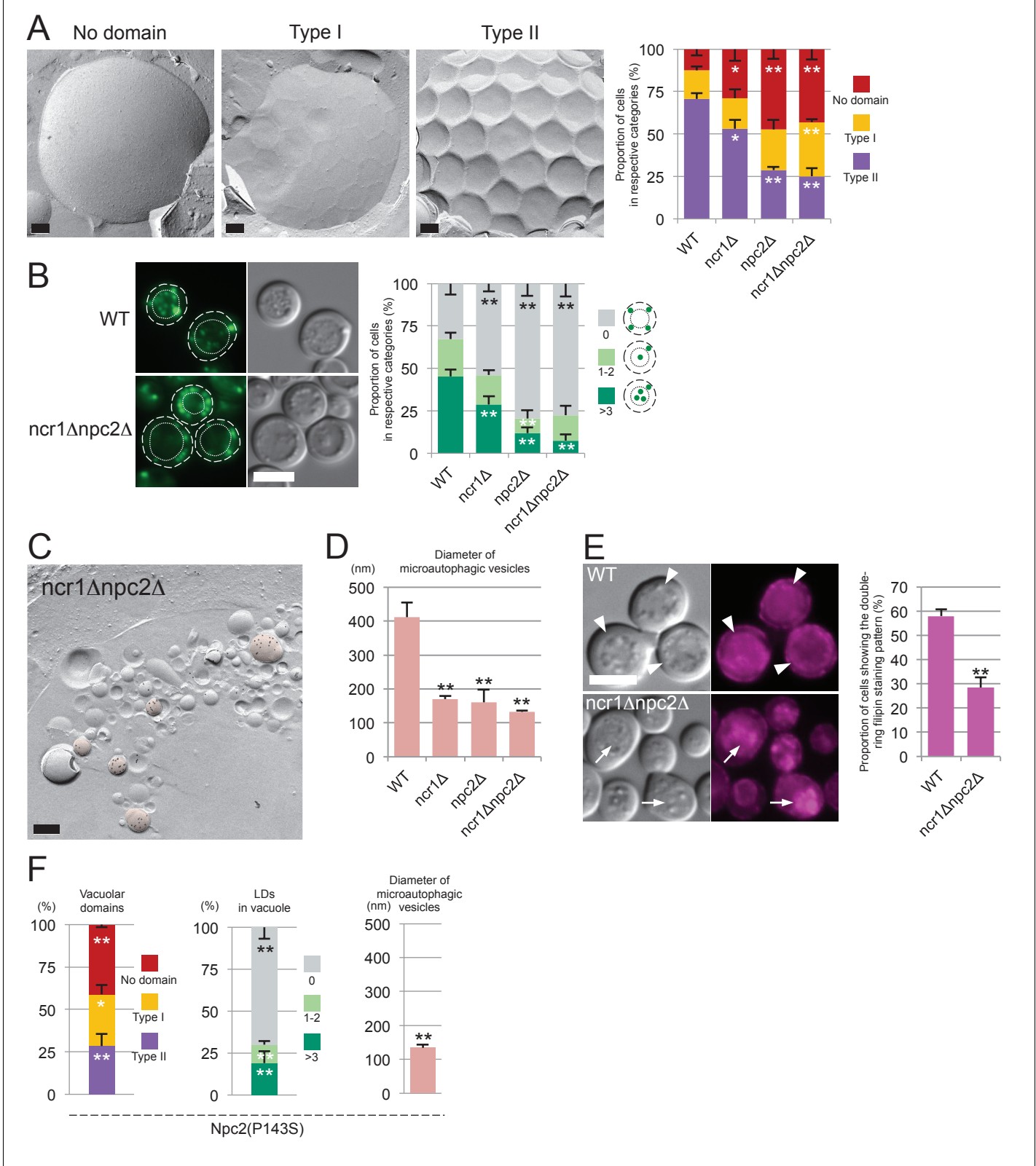

**Figure 3.** Sterol transport by NPC proteins is essential for stationary phase lipophagy. (**A**) Vacuolar domains were classified in three categories: no domain, domains without inward curvature (type I), and domains with inward curvature (type II). Bars: 0.2 µm. In comparison to WT, the type II vacuoles were significantly decreased in NPC-deficient cells. Mean ± SD of three independent experiments (>50 vacuoles were counted for each group in each *Figure 3 continued on next page*

*Figure 3 continued*

experiment). *p<0.05; **p<0.01 (B) Lipophagy in stationary phase. LDs were stained with BODIPY493/503. Bar: 5 µm. The vacuoles were classified by the number of LDs in the lumen: that is, no LD, 1–2 LDs, and more than 3 LDs. There were significantly fewer vacuoles containing LDs in NPC-deficient cells than in WT. Mean ± SD of a single representative experiment (n > 150 for each group) out of three repeated experiments. **p<0.01. (C) Intravacuolar vesicles. Microautophagic vesicles (defined by PtdIns[3]P labeling of more than 100 vesicles/µm$^2$ in the cytoplasmic leaflet; colored in red) were small in *ncr1△ npc2△* (see *Figure 2B* for comparison with WT). Many small vesicles with low levels of PtdIns(3)P labeling were also observed. See *Figure 3—figure supplement 2* for *ncr1△* and *npc2△*. Bar: 0.2 µm. (D) Sizes of microautophagic vesicles. Mean ± SD of three independent experiments. See *Figure 3—figure supplement 2B* for a box plot. **p<0.01. (E) Filipin staining. The proportion of cells showing the double-ring filipin-staining pattern was significantly lower in NPC-deficient cells than in WT [mean ± SD of three independent samples (>150 vacuoles each were counted for each group)]. The vacuole of NPC-deficient cells showed prominent sterol deposition in the vacuolar lumen (arrows). Bar: 5 µm. **p<0.01. (F) Yeast expressing Npc2p(P143S) showed all the defects observed in *npc2△*. The domain formation, lipophagy, and the size of the microautophagic vesicles in Npc2p(P143S) were compared with those in WT as shown in *Figure 3A,B and D*, respectively. *p<0.05; **p<0.01

The following figure supplements are available for figure 3:

**Figure supplement 1.** Filipin staining.

**Figure supplement 2.** The number and sizes of LDs in stationary phase.

**Figure supplement 3.** Microautophagic vesicles in the lumen of a stationary phase vacuole.

Third, the vesicles in the vacuole were examined by freeze-fracture EM using PtdIns(3)P to define microautophagic vesicles (*Figure 3C* and *Figure 3—figure supplement 3A*). The size of microautophagic vesicles was significantly smaller in *ncr1△*, *npc2△*, and *ncr1△npc2△* than in WT (*Figure 3D* and *Figure 3—figure supplement 3B*). Those small microautophagic vesicles were not likely to accommodate LDs, which have diameters ranging from 0.3 µm to 0.5 µm both in WT and in NPC-deficient cells. The other prominent feature of NPC-deficient vacuoles was the presence of a large number of small PtdIns(3)P-deficient vesicles (*Figure 3C* and *Figure 3—figure supplement 3A*).

Fourth, the proportion of cells showing the double-ring filipin-staining pattern was significantly lower in NPC-deficient cells than in WT cells, indicating that sterol transport to the vacuolar membrane may be suppressed in NPC-deficient cells (*Figure 3E*). Moreover, NPC-deficient cells showed deposition of sterol-rich materials in the vacuole lumen (*Figure 3E*), which most probably corresponds to the small PtdIns(3)P-deficient vesicles observed by freeze-fracture EM (*Figure 3C* and *Figure 3—figure supplement 3A*). The paucity of PtdIns(3)P labeling in those vesicles suggested that they were neither microautophagic vesicles nor macroautophagic bodies per se.

In all of the experiments described above, *npc2△* showed a stronger phenotype than *ncr1△*. Moreover, a single amino acid substitution in Npc2p, that is, replacement of the 143$^{rd}$ proline with serine (P143S), the mutation that abrogates cholesterol binding in the corresponding residue of human NPC2 (P120S) (*Wang et al., 2010*), recapitulated all the defects of NPC-deficient cells (*Figure 3F* and *Figure 3—figure supplement 3B*). These results indicated that NPC proteins, Npc2p in particular, promote the formation and expansion of the vacuolar raft-like domain, causing stationary phase lipophagy.

## Trafficking defect of NPC proteins impairs stationary phase lipophagy

Vacuolar domain formation and/or lipophagy in stationary phase were reported to be abrogated in several gene-deletion mutants, but the mechanism underlying the defect was not necessarily clear (*Toulmay and Prinz, 2013*; *Wang et al., 2014*). We suspected that some of the gene deletions may cause abnormalities by altering the distribution of NPC proteins.

In WT, Ncr1p–GFP and Npc2p–GFP were observed in the membrane and the lumen of log-phase vacuoles, respectively (*Berger et al., 2005*; *Zhang et al., 2004*), and a similar distribution was preserved in stationary phase vacuoles (*Figure 4A*). In cells lacking a core *atg* gene, i.e., *atg1△*, *atg2△*, *atg3△*, *atg5△*, *atg7△*, *atg8△*, and *atg18△*, Ncr1p–GFP and Npc2p–GFP were seen in the vacuole in log phase, but they were occasionally observed in puncta adjacent to the vacuole in post-diauxic phase, and finally, in stationary phase, they were found only in the puncta in a majority of cells (*Figure 4A* and *Figure 4—figure supplement 1*). The identity of the NPC puncta is not clear at

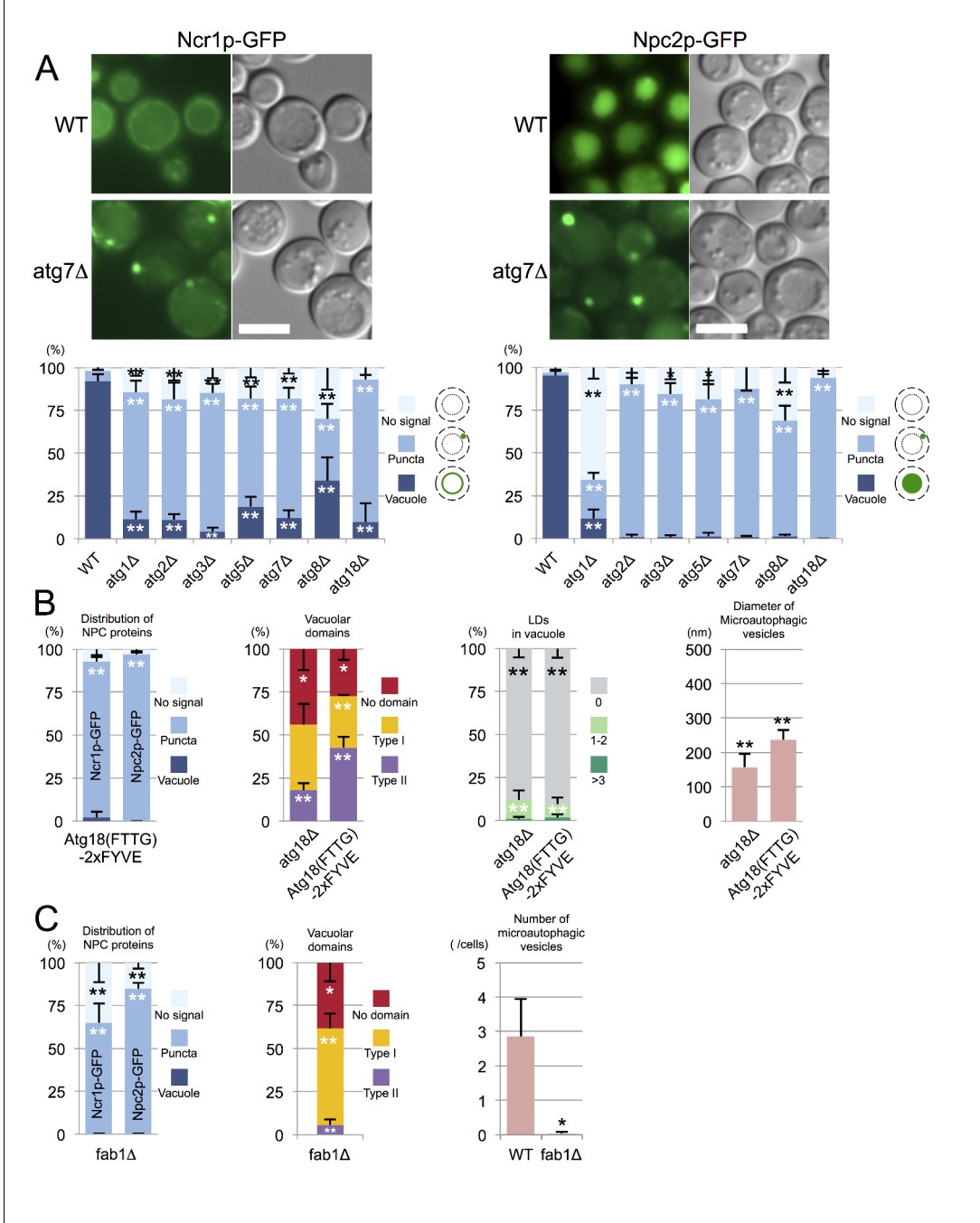

**Figure 4.** Trafficking defect of NPC proteins impairs stationary phase lipophagy. (**A**) Ncr1p–GFP and Npc2p–GFP in stationary phase were targeted to the vacuole in WT, but showed punctate distribution in a majority of Atg-deficient cells. Bar: 5 μm. The quantification data show the mean ± SD of a single representative experiment (n > 150 for each group) out of two to three replicates. (**B**) Expression of Atg18(FTTG)−2xFYVE did not correct the defects in *atg18△* in stationary phase. Aberrant NPC distribution, decreased vacuolar domain formation, decreased lipophagy, and smaller microautophagic vesicles were observed in *atg18△* expressing Atg18(FTTG)−2xFYVE. Respective data were obtained as in *Figures 4A*, *3A, B and D*. (**C**) *fab1△* showed aberrant NPC distribution, decreased vacuolar domain formation, and virtually no microautophagic vesicles. The data on NPC proteins and domain formation were obtained as in *Figures 4A* and *3A*. The number of microautophagic vesicles was counted in three independent experiments. Mean ± SD (n = 3; >15 vacuoles were counted in each experiment). *p<0.05; **p<0.01.

The following figure supplements are available for figure 4:

**Figure supplement 1.** Time course of Ncr1p–GFP distributional change in WT and mutants.

*Figure 4 continued on next page*

*Figure 4 continued*

**Figure supplement 2.** Distribution of Vph1p–mRFP.
**Figure supplement 3.** Filipin staining of *atg7Δ* in stationary phase.
**Figure supplement 4.** Microautophagic vesicles in *atg18Δ* with or without expressing Atg18(FTTG)−2xFYVE in stationary phase.
**Figure supplement 5.** Freeze-fracture replica EM of *fab1Δ* in stationary phase.
**Figure supplement 6.** Distribution of Ncr1p–GFP and Npc2p–GFP in stationary phase.

present, but their aberrant distribution was not likely to be caused by total disorder of protein transport, because Vph1p–mRFP showed normal vacuolar distribution (*Figure 4—figure supplement 2*). Consistent with the decrease in NPC proteins in the vacuole, the proportion of cells showing the double-ring filipin-staining pattern was lower in *atg7△* than in WT (*Figure 4—figure supplement 3*).

To examine whether the aberrant distribution of NPC proteins in *atg* mutants occurred as a result of macroautophagy deficiency, we focused on *atg18△*. Atg18p has two different functionalities, one related to autophagosome biogenesis (*Barth et al., 2001*) and the other related to the regulation of vacuolar morphology (*Efe et al., 2007*). These two functions require binding to PtdIns(3)P and PtdIns(3,5)$P_2$, respectively, and the expression of Atg18(FTTG)−2xFYVE, which binds to PtdIns(3)P but not to PtdIns(3,5)$P_2$, restores macroautophagy in *atg18△* (*Obara et al., 2008*). Even in *atg18△* expressing Atg18(FTTG)−2xFYVE, however, NPC proteins in stationary phase showed abnormal distribution, and domain formation and lipophagy, and the size of the microautophagic vesicles remained significantly different from those in WT (*Figure 4B* and *Figure 4—figure supplement 4*). Additionally, in *pep4△*, which is defective in the recycling of autophagocytosed materials, NPC proteins were present in the vacuole even in stationary phase (*Figure 4—figure supplement 1*). These results indicated that the restoration of macroautophagy is not sufficient to normalize stationary phase lipophagy in *atg18△* and that PtdIns(3,5)$P_2$-dependent Atg18p functionality may be required. Yet this does not rule out the possibility that the macroautophagy mechanism is involved in stationary phase lipophagy.

The importance of PtdIns(3,5)$P_2$ was further confirmed by the observation of PtdIns(3,5)$P_2$-deficient *fab1△*, in which the punctate distribution of NPC proteins was already prevalent in post-diauxic phase and the membrane domain formation was reduced (*Figure 4C* and *Figure 4—figure supplement 1*). The complete absence of vesicles in the vacuolar lumen of *fab1△* suggested that defects besides abnormal NPC trafficking may also exist in this mutant (*Figure 4C* and *Figure 4—figure supplement 5*).

Among the other mutants that were defective in domain formation (*Toulmay and Prinz, 2013*), *vps4△* and *nem1△* showed the aberrant dot distribution of NPC proteins (*Figure 4—figure supplement 6*) (other defects in *vps4△* will be shown later). In these as well as in *atg* mutants, the mechanism causing the abnormal distribution of NPC proteins in stationary phase is not clear and demands further study.

## Microautophagy in acute nitrogen starvation also occurs in NPC-dependent raft-like domains

In light of the findings in stationary phase, we hypothesized that sterol transport by NPC proteins may also be involved in microautophagy in acute nitrogen starvation (*Roberts et al., 2003*; *van Zutphen et al., 2014*). In fact, IMP-deficient domains were found in the vacuole of yeast cultured in nitrogen-deficient medium (*Figure 5A*), and exclusion of Vph1p labeling (*Figure 5—figure supplement 1*) from those domains indicated that they also have raft-like properties. Vacuoles often harbored several raft-like domains, but vacuoles that were entirely covered with them, as in stationary phase, were scarce.

In comparison to WT, NPC-deficient cells in nitrogen starvation had far fewer raft-like domains (*Figure 5B*), and this reduction in domain formation was more prominent in *npc2△* than in *ncr1△*. Filipin staining suggested a relative increase in sterol in the vacuolar membrane of WT, whereas

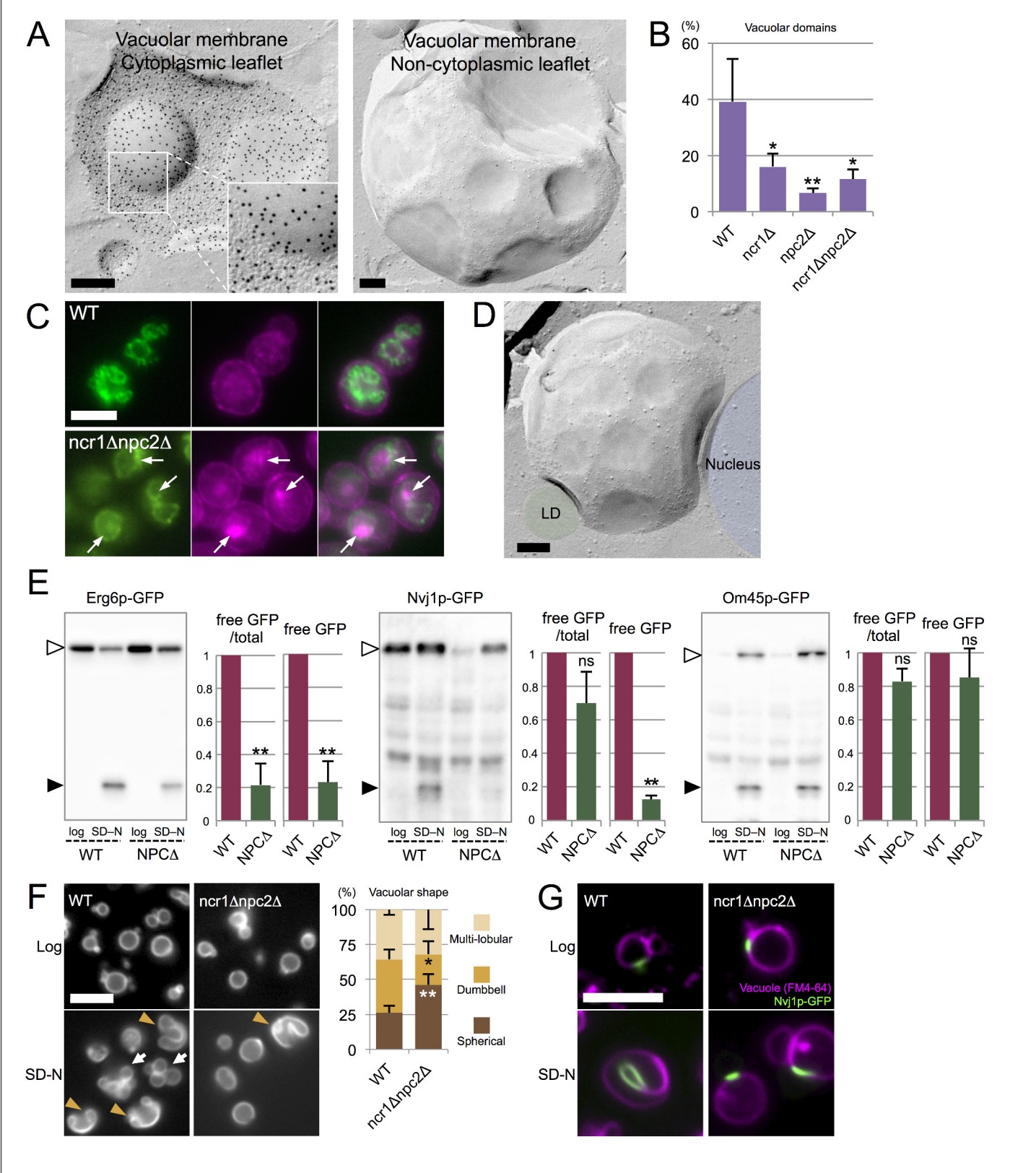

**Figure 5.** Microautophagy in acute nitrogen starvation also occurs at NPC-dependent raft-like domains. (**A**) The IMP-deficient raft-like domains in the vacuolar membrane of WT after nitrogen starvation for 5 hr. These domains were observed both in the cytoplasmic leaflet (P face) and the non-cytoplasmic leaflet (E face). The P face is densely labeled for PtdIns(3)P both in the IMP-rich and IMP-deficient domains. Bars: 0.2 μm. (**B**) IMP-deficient

*Figure 5 continued on next page*

Figure 5 continued

raft-like domains were significantly less frequent in NPC-deficient cells than in WT. Cells showing more than one raft-like domain were counted. Mean ± SD (n > 3; >50 vacuoles were counted for each group in each experiment). *p<0.05; **p<0.01. (C) Filipin staining in nitrogen-starved cells. Sterol (magenta) was stained in the area overlapping with the vacuolar membrane visualized by Vph1p–GFP (green) in WT, whereas in ncr1△npc2△, sterol-rich deposits (arrows) were observed in the vacuolar lumen encircled by Vph1p–GFP. Bar: 5 μm. (D) The IMP-deficient domain showed close apposition to an LD (green) and the nucleus (blue). Bar: 0.2 μm. See also *Figure 5—figure supplement 2*. (E) Degradation of organelle markers by nitrogen starvation for 5 hr: Erg6p–GFP (LD), Nvj1p–GFP (nucleus), and Om45p–GFP (mitochondria). The intensity of free GFP bands (black arrowheads) and the relative ratio of free GFP to the full-length protein (white arrowheads) were measured. Degradation of Erg6p–GFP and Nvj1p–GFP was significantly reduced in ncr1△npc2△ (NPC△) compared to WT, whereas degradation of Om45p–GFP was not. Mean ± SEM (n = 3). **p<0.01. (F) Vacuoles (Vph1p–mRFP) showed dumbbell-like (arrowheads) and multi-lobular shapes (arrows) after nitrogen starvation in WT, but shape change was much less frequent in ncr1△npc2△. Bar: 5 μm. Mean ± SD of a single representative experiment (n > 150 for each group) out of three repeats. *p<0.05; **p<0.01. (G) Nvj1p–GFP (green) in WT showed marked elongation along the deformed vacuole after nitrogen starvation (magenta; FM4-64), but this change was minimal in ncr1△npc2△. Bar, 5 μm.

The following figure supplements are available for figure 5:

**Figure supplement 1.** Distribution of Vph1p–mRFP in the vacuolar membrane after nitrogen starvation.
**Figure supplement 2.** Close association of the raft-like domain and organelles.
**Figure supplement 3.** Effects of NPC deficiency in nitrogen starvation.

sterol deposition in the vacuolar lumen was prominent in ncr1△npc2△ (*Figure 5C*), indicating that NPC proteins are also engaged in sterol transport in nitrogen starvation.

Furthermore, in WT after nitrogen starvation, LDs and the nucleus were frequently associated with the vacuoles, and their association always occurred in the raft-like domain (*Figure 5D* and *Figure 5—figure supplement 2*). LDs were often found within deep vacuolar invaginations, suggesting that they were processed by a microautophagic mechanism similar to that in stationary phase. The LD-containing invagination was morphologically different from the autophagic tube, which is a tubular extension with an IMP-deficient vesicle at the tip (*Müller et al., 2000*).

To determine whether NPC proteins are necessary for microautophagy in nitrogen starvation, we examined the degradation of two marker proteins, Erg6p–GFP for LDs and Nvj1p–GFP for the nucleus, by Western blotting. After nitrogen starvation, the degradation of Erg6p–GFP was significantly less advanced in ncr1△npc2△ than in WT, as shown both by the amount of free GFP and by the relative proportion of free GFP in the total protein amount (*Figure 5E*). We confirmed that the number and size of LDs are not drastically different between WT and NPC-deficient cells (*Figure 5—figure supplement 3A*). Nvj1p–GFP also showed significantly less generation of free GFP in ncr1△npc2△ than in WT (*Figure 5E*), but the difference in the proportion of free GFP in the total protein was not significant between WT and ncr1△npc2△. The latter result was apparently produced because the quantity of full-length Nvj1p–GFP in WT increased after nitrogen starvation, reflecting the expansion of the nuclear-vacuolar junction (NVJ) (see *Figure 5G*).

By contrast, the degradation of the mitochondrial marker Om45p–GFP (*Figure 5E*) and the bulk autophagy activity measured by the Pho8Δ60 assay (*Figure 5—figure supplement 3B*) were at comparable levels in WT and NPC-deficient cells. In conjunction with the EM data, this finding confirmed that NPC proteins play a critical role specifically in microautophagy of LDs and the nucleus.

Vacuoles were largely spherical in log phase both in WT and in NPC-deficient cells, but under nitrogen starvation a majority of the vacuoles showed gross deformation, taking multi-lobular or dumbbell shapes, only in WT (*Figure 5F*). This vacuolar deformation made it difficult to capture microautophagy using light microscopy, but LDs were often observed along the indented surface of the dumbbell-shaped vacuoles, indicative of their close association (*Figure 5—figure supplement 3C*). Notably, NVJs that were marked by Nvj1p–GFP in WT was changed from a short line in log phase to a longer line along a deformed vacuole after nitrogen starvation, whereas such a change was negligible in ncr1△npc2△ (*Figure 5G*).

## The MVB pathway supplies sterol for the induction of microautophagy

We next asked about the source of sterol that is transported by NPC proteins within the vacuole. To investigate this matter, we examined the identity of the sterol-rich material observed in the vacuolar lumen of *ncr1△npc2△* (*Figure 5C*). Freeze-fracture EM revealed that smooth vesicles and small rugged structures were constantly present in the vacuolar lumen in *ncr1△npc2△* but only scarcely in WT (*Figure 6A,B*). The smooth vesicles were often labeled for PtdIns(3)P in the cytoplasmic leaflet, indicating that they are microautophagic vesicles. On the other hand, the rugged structure was morphologically identified as the intraluminal vesicle (ILV) of the multi-vesicular body (MVB) (*Figure 6—figure supplement 1A*). Previously, mammalian ILVs have been shown to be enriched with sterol (*Möbius et al., 2002*), and thus we hypothesized that ILVs in yeast may be the major source of sterol to be transported by NPC proteins in nitrogen starvation.

Three lines of evidence support this idea. First, in *vps4△*, in which the MVB pathway is blocked, hardly any degradation of Erg6p–GFP and Nvj1–GFP was observed (*Figure 6C*) and the number of raft-like domains in the vacuolar membrane was significantly decreased (*Figure 6D*). Consistently, filipin failed to stain the vacuolar membrane in *vps4△* after nitrogen starvation (*Figure 6—figure supplement 1B*). Here it is of note that, in *vps4△*, macroautophagy is not suppressed (*Babst et al., 1997*) and NPC proteins show normal vacuolar distribution (*Figure 6—figure supplement 1C*). Second, in cells in which two major vacuolar proteases (*pep4△prb1△*) are depleted, the vacuolar lumen harbored a large number of ILVs as well as macroautophagic bodies (*Figure 6E*), and the raft-like domain in the vacuolar membrane was reduced to a minimum level (*Figure 6D*). This result, together with the persistence of ILVs and microautophagic vesicles in *ncr1△npc2△* (*Figure 6A,B*), suggested that both proteolysis and NPC-mediated sterol extraction are necessary for efficient degradation of sterol-rich membranes such as ILVs. Finally, to confirm that yeast ILVs are enriched with sterol, *atg7△pep4△prb1△* was examined after nitrogen starvation. As expected from the results from *pep4△prb1△*, the vacuole of *atg7△pep4△prb1△* contained abundant ILVs with few other structures, whereas that of *atg7△* was largely vacant (*Figure 6—figure supplement 1D*). Filipin staining yielded a diffuse fluorescence signal in the vacuolar lumen of *atg7△pep4△prb1△*, but not of *atg7△* (*Figure 6F*). The combined result indicated that ILVs are enriched with sterol. On the basis of these results, we concluded that sterol brought into the vacuole by ILVs is crucial for the formation and expansion of the raft-like domain and microautophagy in nitrogen starvation.

The above result prompted us to ask whether ILVs also play a role in the domain formation in the stationary phase vacuole. In stationary phase, sterol esters in LDs were proposed to be a sterol source (*Wang et al., 2014*). Consistently, raft-like domain formation and lipophagy were decreased to low levels in *are1△are2△* that cannot synthesize sterol esters (*Figure 6—figure supplement 2*). On the other hand, domain formation and lipophagy in the stationary phase vacuoles were absent in *vps4△* (*Figure 6—figure supplement 1E*). Abnormal trafficking of NPC proteins may be a cause of the defect, but the complete absence of the raft-like domain in *vps4△* suggests that sterol in ILVs may be important as an initial trigger to induce the stationary phase vacuolar domain.

## Involvement of Atg proteins and actin filaments in microautophagy during nitrogen starvation

Microautophagy of LDs and the nucleus in nitrogen starvation is suppressed in *atg* mutants (*Roberts et al., 2003*; *van Zutphen et al., 2014*). We confirmed significant reduction in the degradation of Erg6p–GFP and Nvj1p–GFP in *atg* mutants, and wanted to determine the point in the process at which microautophagy is aborted in these cells. The vacuolar distribution of Ncr1p–GFP and Npc2p–GFP in *atg* mutants was preserved in acute nitrogen starvation (*Figure 7—figure supplement 1*), but the numbers of raft-like domains were significantly reduced (*Figure 7A*). The activation of the MVB pathway in nitrogen starvation has been reported to occur similarly even in the absence of macroautophagy, suggesting that ILVs should reach the vacuolar lumen (*Müller et al., 2015*). Nevertheless, few structures were observed in the vacuolar lumen of *atg* mutants (*Figure 6—figure supplement 1D*), suggesting that ILV degradation and sterol transport to the vacuolar membrane may be occurring normally. This result inferred that raft-like domain formation in nitrogen starvation was suppressed in *atg* mutants even though sterol is transported to the vacuolar membrane as in WT (see Discussion for the possible cause of this defect).

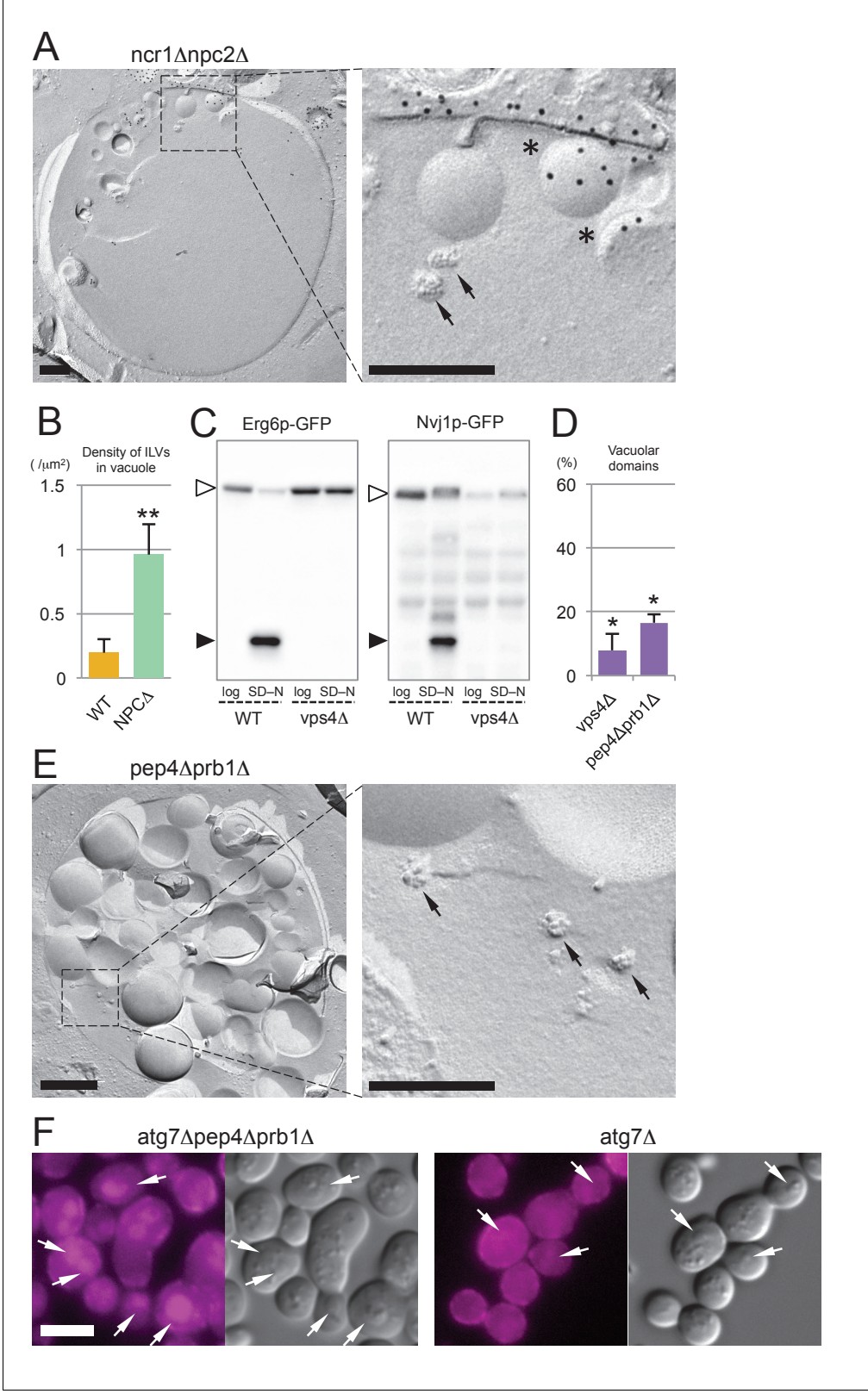

**Figure 6.** The MVB pathway supplies sterol for induction of microautophagy. (**A**) The vacuolar content of *ncr1△npc2△* after nitrogen starvation for 5 hr. ILVs, seen as small rugged structures (arrows), and microautophagic vesicles labeled for PtdIns(3)P in the cytoplasmic leaflet (*) were observed. Bars: 0.2 μm. (**B**) The density of ILVs in the vacuolar lumen after nitrogen starvation was significantly higher in *ncr1△npc2△* than in WT. Mean ±

*Figure 6 continued on next page*

*Figure 6 continued*

SD are represented (n = 3; >30 vacuoles were counted in each experiment). **p<0.01. (**C**) Degradation of Erg6p–GFP and Nvj1p–GFP in nitrogen starvation was suppressed in *vps4△*. The bands of the full-length proteins (white arrowheads) and free GFP (black arrowheads) are marked. (**D**) Raft-like domains in the vacuolar membrane were significantly less frequent in *vps4△* and *pep4△prb1△* than in WT (see *Figure 5B* for WT). *p<0.05. (**E**) The vacuolar content of *pep4△prb1△* after nitrogen starvation. The vacuolar lumen was filled with autophagic bodies (left figure; bar, 0.5 µm) and ILVs (arrows in the right panel; bar, 0.2 µm). (**F**) Filipin staining showed that the vacuolar lumen of *atg7△pep4△prb1△* after nitrogen starvation contains abundant sterol-rich structures, whereas that of *atg7△* does not (arrows). Combined with the result of freeze-fracture EM (see *Figure 6—figure supplement 1D*), the filipin fluorescence in the vacuolar lumen of *atg7△pep4△prb1△* is thought to be derived from ILVs. Bar: 5 µm.

The following figure supplements are available for figure 6:

**Figure supplement 1.** Analysis of MVB and *vps4Δ*.

**Figure supplement 2.** *are1Δ are2Δ* in stationary phase.

To further characterize microautophagy in nitrogen starvation, actin filaments and microtubules were depolymerized by latrunculin A and nocodazole, respectively, and the effect was examined. Latrunculin A suppressed degradation of both Erg6p–GFP and Nvj1p–GFP, whereas nocodazole did not (*Figure 7B*). The larunculin A treatment also decreased the number of raft-like domains in the vacuolar membrane in WT, but it did not affect the already reduced number of raft-like domains in *ncr1△npc2△* (*Figure 7C*). On the other hand, the gross shape change in vacuoles was reduced in WT treated with latrunculin A and further decreased in *ncr1△npc2△* (*Figure 7D*), indicating that the effects of actin depolymerization and NPC deletion were additive in this respect. This result suggested that actin depolymerization may have an additional effect besides suppressing raft-like domain formation.

On the basis of all of these results, we propose the mechanism for microautophagy in nitrogen starvation, as depicted in *Figure 7E*.

## Discussion

### NPC proteins are essential for the expansion of raft-like domains and microautophagy

The absence of Ncr1p or Npc2p does not cause overt abnormalities in yeast cells cultured under normal growth conditions (*Berger et al., 2005*; *Malathi et al., 2004*; *Zhang et al., 2004*). This may be because the yeast vacuole does not need to process endocytosed lipoproteins as mammalian cells do, but this possible explanation raises the question of which functions NPC proteins have in yeast. The present study showed that NPC proteins play a crucial role in the expansion of raft-like vacuolar membrane domains in microautophagy, probably by transporting sterol to the vacuolar membrane.

Obviously, NPC proteins are not the only mechanism capable of transporting sterol to the vacuolar membrane. This is indicated by the presence of raft-like domains in NPC-deficient cells, albeit at a lower level than in WT. Nevertheless, stationary phase lipophagy is prominently suppressed in NPC-deficient cells. This result suggests that sterol transport by NPC proteins is important not just for raft-like domain formationbut more critically so for the expansion of those domains, and this latter process appears indispensable for the engulfment of cargo in microautophagy.

How can NPC proteins expand the raft-like domain and promote microautophagy? In this context, it is notable that depletion of Npc2p causes more significant defects in microautophagy than does depletion of Ncr1. The Npc2p dominance indicates that direct sterol transport by Npc2p (bypassing Ncr1p), as shown for mammalian NPC proteins (*Kennedy et al., 2012*; *Xu et al., 2008*), may play a major role in the expansion of raft-like domains. Npc2p was shown to bind preferentially with negative-charged membranes (*McCauliff et al., 2015*), and thus the sphingolipid-enriched raft-like domain is a suitable platform for the interaction with Npc2p. Hence, it is plausible that

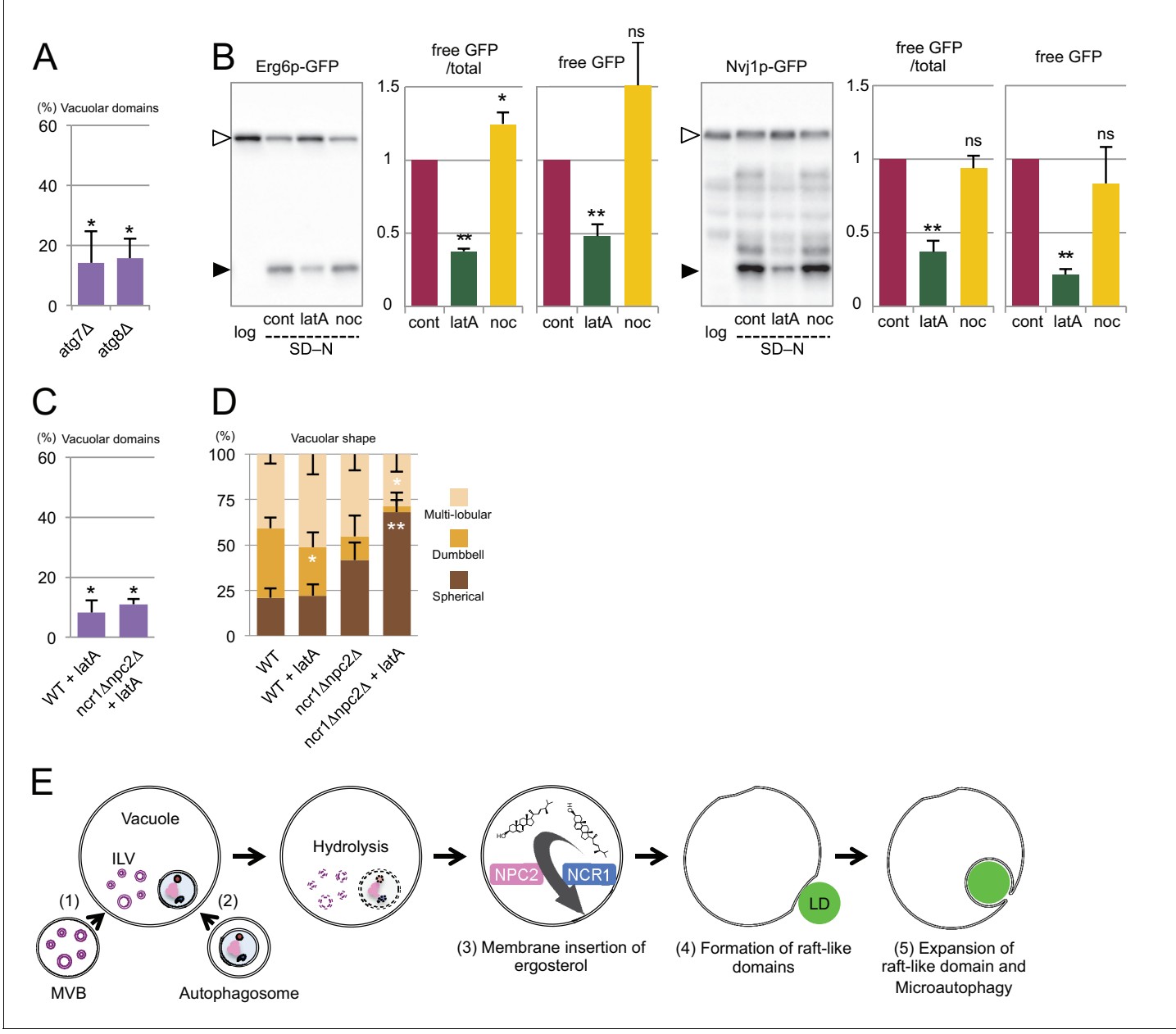

**Figure 7.** Involvement of *Atg* proteins and actin filaments in microautophagy during acute nitrogen starvation. (**A**) Raft-like domain formation after nitrogen starvation was significantly less frequent in *atg7Δ* and *atg8Δ* than in WT (see *Figure 5B* for WT). (**B**) Degradation of Erg6p–GFP and Nvj1p–GFP in nitrogen starvation was suppressed by latrunculin A (0.2 mM; latA), but not by nocodazole (10 µg/ml; noc). Mean ± SEM (n = 3) are shown. The intensity of free GFP bands (black arrowheads) and the relative density ratio of free GFP to the full-length protein (white arrowheads) were quantified. *p<0.05; **p<0.01. (**C**) Raft-like domain formation after nitrogen starvation in WT was reduced by latrunculin A (latA). Raft-like domain in *ncr1Δnpc2Δ*, which was already low, was not further decreased by latrunculin A. The result was compared to that of WT and *ncr1Δnpc2Δ* shown in *Figure 5B*. (**D**) The rate of vacuolar shape change in nitrogen starvation was reduced by latrunculin A (latA). Vacuoles were classified as in *Figure 5F*. Latrunculin A further decreased the rate of shape change in *ncr1Δnpc2Δ*. The results (mean ± SD) of one representative experiment (n > 150 for each group) out of three repeats are shown. *p<0.05; **p<0.01. (**E**) The putative mechanism of microautophagy in nitrogen starvation. (1) The activated MVB pathway delivers sterol-rich ILVs to the vacuole; (2) autophagosomes fuse with the vacuolar membrane to supply sphingolipids; (3) NPC proteins transport sterol from decomposing ILVs to the vacuolar membrane; (4) the generated raft-like domain adheres to cargo organelles via raft-philic receptors; and (5) the raft-like domain expands to engulf cargos leading to microautophagy.

The following figure supplement is available for figure 7:

**Figure supplement 1.** Distribution of Ncr1p–GFP and Npc2p–GFP after nitrogen starvation.

recruitment of Npc2p to the raft-like domain facilitates the domain's expansion through direct sterol transfer. Exclusion of Ncr1p from raft-like domains is also consistent with this deduction (*Toulmay and Prinz, 2013*).

The other important finding that contributes to our understanding of the mechanism of NPC proteins' functioning is the manner in which raft-like domains invaginate. Inward budding of raft-like domains has been observed in giant liposomes (*Hamada et al., 2007*), suggesting that it may be an endogenous propensity of those domains. However, the vacuolar invagination differs from the liposomal phenomenon in that it occurs only in limited raft-like domains. This was particularly obvious in stationary phase vacuoles, where polygonal raft-like domains exist over the entire vacuole surface but invagination occurs only in the domains adhering to cargo. This result suggests that tight adhesion to cargo is conducive to invagination. We speculate that the inward curvature imposed on raft-like domains by the adhesion may create packing defects in the luminal leaflet of the vacuolar membrane (*Cui et al., 2011*), thereby enhancing the Npc2p-mediated sterol transfer.

Finally, it is noteworthy that Vac8p, the vacuolar protein interacting with Nvj1p, has an affinity to rafts (*Peng et al., 2006*). We hypothesize that this raft-philicity of vacuolar interacting molecules is important for the microautophagic process,because enlargement of the raft-like domain and recruitment of additional interacting proteins may proceed in a feed-forward manner. The marked elongation of NVJ in nitrogen starvation is consistent with this idea.

## The MVB pathway supplies sterol for the induction of microautophagy

The MVB pathway and endocytic degradation of plasmalemmal proteins are activated immediately after amino acid starvation; and the resultant proteome remodeling, including upregulation of vacuolar hydrolases, is thought to boost macroautophagic degradation (*Jones et al., 2012*; *Müller et al., 2015*). The present study showed that this activation of the MVB pathway is also important for microautophagy and that ILVs are a major source of sterol that is transported by NPC proteins. Hence, in acute nitrogen starvation, both macroautophagy and microautophagy appear to be coordinated with the MVB pathway to support cell survival.

The importance of the MVB pathway in microautophagy in the stationary phase vacuole is less clear than that in acute nitrogen starvation because of the longer time course. Nevertheless, we think that as cells enter the post-diauxic phase, the MVB pathway is likely to be upregulated because the recycling of endocytosed plasmalemmal proteins from the endosome to the plasma membrane is reduced in glucose starvation (*Lang et al., 2014*). Complete lack of raft-like domains in *vps4△*, in which ILVs do not form, is consistent with this assumption. We thus propose that the MVB pathway may be involved in an early stage of raft-like domain formation in the stationary phase vacuole, whereas LDs may become an important sterol source at later time points after the MVB pathway subsides.

## Correlation of macroautophagy and microautophagy in nitrogen starvation

Microautophagy in acute nitrogen starvation is suppressed in *atg* mutants (*Roberts et al., 2003*; *van Zutphen et al., 2014*), but how this defect is correlated with macroautophagy deficiency has not been clear. We found that the raft-like domain formation in nitrogen starvation is significantly down-regulated in *atg* mutants. This is peculiar because the activation of the MVB pathway (*Müller et al., 2015*) and the vacuolar distribution of NPC proteins are observed in *atg* mutants, suggesting that sterol transport to the vacuolar membrane occurs in these mutants on a level comparable to that in WT.

We speculate that the defect of microautophagy in *atg* mutants may be caused by deficiency of sphingolipids, the other indispensable component of raft-like domains (*Lingwood and Simons, 2010*). Considering that the vacuolar membrane is relatively poor in sphingolipids under normal growing conditions (*Hechtberger et al., 1994*; *Schneiter et al., 1999*), sphingolipids are thought to be imported to the vacuole to form the raft-like domains in nitrogen starvation. Fusion of autophagosomes is the most likely route for such transport (*Yamagata et al., 2011*), and this explains why *atg* mutants that are deficient in autophagy show few raft-like domains. Nevertheless, the possibility that other sphingolipid transport pathways that are downregulated in *atg* mutants may be involved cannot be excluded.

Additional defects may also occur in *atg* mutants after nitrogen starvation. In this context, it is notable that microautophagy in nitrogen starvation was suppressed by depolymerization of actin. In light of the necessity of actin for selective macroautophagy, but not for bulk autophagy (*Reggiori et al., 2005*), one role of actin may be to facilitate the engagement of cargo organelles with the vacuolar membrane. Yet we found that actin depolymerization also decreases raft-like domain formation, which implies that actin has some direct effect on the vacuolar membrane. Considering that actin integrity is important in organizing plasma membrane domains in mammalian cells (*Goswami et al., 2008*; *Suzuki et al., 2007*), actin that is associated with the vacuolar membrane (*Eitzen et al., 2002*) may be involved in generating the raft-like domains. This vacuole-associated actin might not be properly organized in *atg* mutants in nitrogen starvation. These speculations need to be tested in further experiments.

### Implications for mammalian cells

In mammalian cells, microautophagy of cytosolic proteins has been reported to occur in the late endosome by an ESCRT-dependent mechanism (*Sahu et al., 2011*), but it is not known whether microautophagy and the MVB pathway are correlated in the same way that we observed in yeast. It is noteworthy, however, that autophagosomes in mammalian cells may often fuse with MVBs to form amphisomes before fusing with lysosomes to form autophagolysosomes, whereas autophagosomes in yeast fuse directly with vacuoles (*Fader and Colombo, 2009*). The amphisome is unique in that it delivers autophagocytosed cytosolic materials and sterol-rich ILVs to the lysosome simultaneously. Moreover, the relative size of an autophagosome (and an amphisome) to a lysosome (or a vacuole) is much larger in mammals than in yeast, indicating that a single autophagosomal fusion would have a greater impact on the mammalian lysosome than on the yeast vacuole. It is thus probable that the mammalian autophago-lysosomal membrane acquires an amount of both sterol and sphingolipids that is sufficient to generate raft-like domains. On the basis of this inference, we would predict that microautophagy in mammalian cells occurs preferentially in lysosomes after fusion with amphisomes.

## Materials and methods

### Yeast

The yeast strains used in this study are based on the parent strain SEY6210 (*Table 1*). Yeast manipulations were performed according to standard protocols (*Kaiser et al., 1994*).

For stationary phase lipophagy, cells were cultured in synthetic complete (SC) medium (0.17% yeast nitrogen base without amino acids and ammonium sulfate [Becton Dickinson, Franklin Lakes, NJ, USA], 0.5% ammonium sulfate, 2% dextrose, and 0.13% dropout mix) for three days starting at $OD_{600\ nm} \approx 0.15$. To induce macroautophagy, cells cultured to $OD_{600\ nm} \approx 1$ in SC medium or YPD medium (1% yeast extract, 2% polypeptone, and 2% dextrose) were washed twice with distilled water and incubated for 5 hr in nitrogen-depleted medium SD(−N) (0.17% yeast nitrogen base without amino acids and ammonium sulfate and 2% dextrose). In some experiments, the SD(−N) medium was added with 0.2 mM latrunculin A (Nacalai Tesque, Kyoto, Japan) or 10 μg/ml nocodazole (Sigma-Aldrich, St. Louis, MO, USA). All cultures were kept at 30°C.

### Western blotting

Yeast cell extracts for Western blotting were prepared using lithium acetate and sodium hydroxide (*Zhang et al., 2011*). The precipitated sample was dissolved in SDS sample buffer (2% SDS, 10% glycerol, 60 mM Tris-HCl, pH 6.8), and after centrifugation, the supernatant was loaded for SDS-PAGE, electrotransferred to PVDF membrane, and subjected to Western blotting using SuperSignal West Dura Extended Duration Substrate (Thermo Fisher, Waltham, MA, USA). Images were captured with a Fusion Solo S instrument (Vilber Lourmat, Eberhardzell, Germany) and analyzed by the accompanying software.

### Fluorescence microscopy

For labeling of LDs, yeast cells were incubated with 0.5 μg/ml BODIPY493/503 (Thermo Fisher) for 10 min, and rinsed with 50 mM Tris-HCl (pH 7.5) immediately before microscopy (*Wang et al.,*

**Table 1.** Yeast strains used in this study.

| Name | Genotype | Reference/origin |
|---|---|---|
| SEY6210 | *MATα leu2-3,112 ura3-52 his3-Δ200 trp1-Δ901 lys2-801 suc2-Δ9* | *Robinson et al. (1988)* |
| KVY4 | SEY6210; *ypt7Δ::LEU2* | *Kihara et al. (2001)* |
| KVY5 | SEY6210; *atg8Δ::HIS3* | *Kirisako et al. (1999)* |
| KVY135 | SEY6210; *atg7Δ::HIS3* | *Obara et al. (2008)* |
| TKY1001 | SEY6210; *atg18Δ::KanMX* | *Cheng et al. (2014)* |
| YT350 | SEY6210; *pep4Δ::hphNT1, prb1Δ::natNT2* | This study |
| YT956 | SEY6210; *fab1Δ::cgTRP1* | This study |
| YT1167 | SEY6210; *NVJ1–yeGFP::klTRP1* | This study |
| YT1242 | SEY6210; *npc2Δ::cgHIS3* | This study |
| YT1276 | SEY6210; *ncr1Δ::cgTRP1, npc2Δ::cgHIS3* | This study |
| YT1278 | SEY6210; *ncr1Δ::cgTRP1, pho8Δ60::natNT2* | This study |
| YT1287 | SEY6210; *VPH1–mRFP::natNT2, ncr1Δ::cgTRP1, npc2Δ::cgHIS3* | This study |
| YT1288 | SEY6210; *ncr1Δ::natNT2* | This study |
| YT1296 | SEY6210; *VPH1–yeGFP::hphNT1* | This study |
| YT1325 | SEY6210; *NCR1–yeGFP::klTRP1* | This study |
| YT1330 | SEY6210; *npc2Δ::cgHIS3, pho8Δ60::natNT2* | This study |
| YT1331 | SEY6210; *ncr1Δ::cgTRP1, npc2Δ::cgHIS3, pho8Δ60::natNT2* | This study |
| YT1349 | SEY6210; *npc2(P143S)::cgTRP1* | This study |
| YT1371 | SEY6210; *NPC2–yeGFP::cgTRP1* | This study |
| YT1373 | SEY6210; *fab1Δ::cgTRP1, NPC2–yeGFP::hphNT1* | This study |
| YT1385 | SEY6210; *atg18Δ::KanMX, NCR1–yeGFP::klTRP1* | This study |
| YT1386 | SEY6210; *atg18Δ::KanMX, NPC2–yeGFP::klTRP1* | This study |
| YT1388 | SEY6210; *fab1Δ::cgTRP1, NCR1–yeGFP::hphNT1* | This study |
| YT1467 | SEY6210; *VPH1–mRFP::natNT2* | This study |
| YT1474 | SEY6210; *nem1Δ::cgHIS3, NCR1–yeGFP::klTRP1* | This study |
| YT1476 | SEY6210; *nem1Δ::cgHIS3, NPC2–yeGFP::klTRP1* | This study |
| YT1479 | SEY6210; *VPH1–yeGFP::hphNT1, ncr1Δ::TRP1, npc2Δ::HIS3* | This study |
| YT1481 | SEY6210; *vps4Δ::cgHIS3, NCR1–yeGFP::klTRP1* | This study |
| YT1482 | SEY6210; *vps4Δ::cgHIS3, NPC2–yeGFP::klTRP1* | This study |
| YT1503 | SEY6210; *atg18Δ::KanMX, ATG18p-atg18(FTTG)–3HA-2FYVE, NCR1–yeGFP::klTRP1* | This study |
| YT1509 | SEY6210; *ncr1Δ::cgTRP1, npc2Δ::cgHIS3, ERG6–yeGFP::klTRP1* | This study |
| YT1517 | SEY6210; *atg18Δ::KanMX, ATG18p-atg18(FTTG)–3HA-2FYVE, NPC2–yeGFP::klTRP1* | This study |
| YT1523 | SEY6210; *ncr1Δ::cgTRP1, npc2Δ::cgHIS3,NVJ1–yeGFP::hphNT1* | This study |
| YT1536 | SEY6210; *ncr1Δ::cgTRP1, npc2Δ::cgHIS3, Om45–yeGFP::hphNT1* | This study |
| YT1542 | SEY6210; *ERG6–yeGFP::natNT2* | This study |
| YT1549 | SEY6210; *vps4Δ::natNT2, NVJ1–yeGFP::hphNT1* | This study |
| YT1552 | SEY6210; *atg7Δ::HIS3, NCR1–yeGFP::klTRP1* | This study |
| YT1553 | SEY6210; *atg3Δ::KanMX, NCR1–yeGFP::klTRP1* | This study |
| YT1554 | SEY6210; *atg5Δ::LEU2, NCR1–yeGFP::klTRP1* | This study |
| YT1556 | SEY6210; *atg2Δ::LEU2, NCR1–yeGFP::klTRP1* | This study |
| YT1557 | SEY6210; *atg1Δ::KanMX, NCR1–yeGFP::klTRP1* | This study |
| YT1558 | SEY6210; *atg7Δ::HIS3, NPC2–yeGFP::klTRP1* | This study |
| YT1559 | SEY6210; *atg3Δ::KanMX, NPC2–yeGFP::klTRP1* | This study |
| YT1560 | SEY6210; *atg5Δ::LEU2, NPC2–yeGFP::klTRP1* | This study |

*Table 1 continued on next page*

*Table 1 continued*

| Name | Genotype | Reference/origin |
|------|----------|------------------|
| YT1561 | SEY6210; atg8Δ::HIS3, NPC2–yeGFP::klTRP1 | This study |
| YT1562 | SEY6210; atg2Δ::LEU2, NPC2–yeGFP::klTRP1 | This study |
| YT1569 | SEY6210; atg1Δ::KanMX, NPC2–yeGFP::klTRP1 | This study |
| YT1579 | SEY6210; atg8Δ::HIS3, NCR1–yeGFP::klTRP1 | This study |
| YT1583 | SEY6210; vps4Δ::cgHIS3, ERG6–yeGFP::klTRP1 | This study |
| YT1584 | SEY6210; atg8Δ::HIS3, NVJ1–yeGFP::klTRP1 | This study |
| YT1586 | SEY6210; atg7Δ::HIS3, ERG6–yeGFP::klTRP1 | This study |
| YT1591 | SEY6210; atg14Δ::LEU2, ERG6–yeGFP::natNT2 | This study |
| YT1592 | SEY6210; atg14Δ::LEU2, NVJ1–yeGFP::natNT2 | This study |
| YT1597 | SEY6210; atg8Δ::HIS3, ERG6–yeGFP::klTRP1 | This study |
| YT1598 | SEY6210; atg7Δ::HIS3, NVJ1–yeGFP::klTRP1 | This study |
| YT1726 | SEY6210; pep4Δ::hphNT1, prb1Δ::natNT2, atg7Δ::HIS3 | This study |

*2014*). The vacuolar membrane was either visualized with Vph1p–mRFP or stained with FM4-64 (Biotium, Fremont, CA, USA) as described previously (*Vida and Emr, 1995*; *Wang et al., 2014*).

For sterol staining, yeast cells were fixed for 30 min with 1% glutaraldehyde in 40 mM potassium phosphate buffer (pH 7) and 0.5 mM magnesium choloride, and treated for 10 min with 0.7% 2-mercaptoethanol in 0.2 M Tris and 20 mM EDTA. After rinses with 0.17 M potassium dihydrogen phosphate and 30 mM sodium citrate (pH 5.8), cells were treated for 5 min with 0.025% Nonidet P-40 in PBS, then with 1 mg/ml sodium borohydride in Tris-buffered saline (pH 8.2), and were labeled for 10 min in 10 μg/ml filipin (Polysciences, Warminster, PA, USA) in PBS. In experiments that looked at filipin staining and Vph1–GFP simultaneously, cells were fixed for 1 hr with 0.25% glutaraldehyde and 5% formaldehyde in 100 mM sodium phosphate buffer (pH 7.4). They were then treated with 0.2 mg/ml Zymolyase 100T (Nacalai Tesque) in PBS and 0.1% 2-mercaptoethanol for 15 min at 30°C before proceeding to the Nonidet P-40 permeabilization, sodium borohydride treatment, and filipin staining.

The stained yeast cells, as well as those expressing GFP- or mRFP-tagged proteins, were mounted on a glass slide and observed under an AxioImager or Axiovert microscope using a 63× NA1.4 Apochromat objective lens (Zeiss, Oberkochen, Germany).

## Quick-freezing and freeze replica preparation

Yeast cells were quick-frozen either by metal-contact freezing using a Slammer (Valiant Instruments, Ellisville, MO, USA) or by high-pressure freezing using an HPM 010 high-pressure freezing machine (Leica Microsystems, Wetzlar, Germany). For metal-contact freezing, cell pellets were placed on a glass coverslip and slammed to a copper block pre-cooled with liquid nitrogen. For high-pressure freezing, an EM grid (200 mesh) impregnated with yeast cells was sandwiched between a 20-μm–thick copper foil and a flat aluminum disc (Engineering Office M. Wohlwend, Sennwald, Switzerland) and frozen according to the manufacture's instructions (*Cheng et al., 2014*).

For freeze-fracture, the specimens were transferred to the cold stage of a Balzers BAF 400 apparatus and fractured at −120 to −100°C under a vacuum of ~$1 \times 10^{-6}$ mbar. Replicas were made by electron-beam evaporation in a process involving three steps: carbon (C) (2–5 nm in thickness) at an angle of 90° to the specimen surface, platinum-carbon (Pt/C) (1–2 nm) at an angle of 45°, and C (10–20 nm) at an angle of 90° (*Fujita et al., 2010*). The thickness of the deposition was adjusted by referring to a crystal thickness monitor. Thawed replicas were treated with 2.5% SDS in 0.1 M Tris-HCl (pH 7.4) at 60°C overnight. To remove the cell wall, yeast replicas were treated for 2 hr at 37°C with one of the following solutions: (1) 20 μg/ml Zymolyase 100T (Nacalai Tesque) in PBS containing 0.1% Triton X-100, 1% BSA, and a protease inhibitor cocktail (Nacalai Tesque) or (2) 0.1% Westase (Takara Bio, Kusatsu, Japan) in McIlvain buffer (37 mM citrate, 126 mM disodium hydrogen phosphate, pH 6.0) containing 10 mM EDTA, 30% fetal calf serum, and a protease inhibitor cocktail. Replicas for

labeling PtdIns(3)P were generally treated with Zymolyase, whereas those for protein labeling were treated with Westase. After the enzyme treatment, the replicas were further treated in 2.5% SDS and stored in buffered 50% glycerol at −20°C until use.

For freeze-fracture/etching, the specimens were fractured at −102°C under a vacuum of ~1 × 10⁻⁶ mbar and kept at the same temperature for 2 min to induce sublimation of water from the fractured surface. The samples were then exposed to rotary evaporation of Pt/C (4 nm) at an angle of 20°, followed by C (20 nm) at an angle of 80°. The replicas were treated with household bleach to digest biological materials before mounting on EM grids for observation. The method of freeze-fracture/etching is described in detail at Bio-protocol (*Tsuji and Fujimoto, 2017*).

### Replica labeling

PtdIns(3)P was labeled as described previously (*Cheng et al., 2014*). Briefly, SDS-treated freeze-fracture replicas were incubated at 4°C overnight with GST-tagged p40$^{phox}$ PX domain (10 ng/ml) in PBS containing 1% BSA, followed by rabbit anti-GST antibody (10 μg/ml) and then by colloidal gold (10 nm)-conjugated protein A (PAG10; University of Utrecht, Utrecht, The Netherlands), each for 30 min at 37°C in 1% BSA in PBST. For labeling of Vph1p–mRFP, replicas were incubated with rabbit anti-mCherry antibody (provided by Dr Shuh-ichi Nishikawa of Niigata University and Dr Masahiko Watanabe of Hokkaido University) followed by PAG10. The labeled replicas were picked up on Formvar-coated EM grids and observed with a JEOL JEM-1011 EM (Tokyo, Japan) operated at 100 kV. Digital images were captured using a CCD camera (Gatan, Pleasanton, CA, USA) and subjected to further analysis.

### Pho8Δ60 assay

Bulk autophagic activity of yeast cells was quantified by measuring phosphatase activity as previously described (*Noda et al., 1995*).

### Statistical analysis

EM images obtained from more than three independent experiments were used for the analysis. The number of colloidal gold particles was counted manually, and areas were measured using ImageJ (NIH). The labeling density in the selected structure was calculated by dividing the number of colloidal gold particles by the area. Statistical differences between samples were examined by Student's t-test except for the results concerning PtdIns(3)P labeling density, the diameter of micro-autophagic vesicles, and Western blotting. Differences in PtdIns(3)P labeling density (*Figure 2C*) and in the diameter of microautophagic vesicles (*Figure 3—figure supplement 3B* and *Figure 4—figure supplement 4B*) were examined by Mann-Whitney's U-test. The Western blotting results (*Figures 5E* and *7B*) were normalized to the controls in their respective blots and subjected to Boot-strapping analysis (*Clèries et al., 2012*). Statistical difference is indicated by *p<0.05 and **p<0.01. The number of technical replicates, that is, the number of multiples of the same sample that were analyzed, is described in respective figure legends.

### Box plots

Box plots were prepared using BoxPlotR (http://boxplot.tyerslab.com/). The center lines show the medians, box boundaries indicate the 25$^{th}$ and 75$^{th}$ percentiles, whiskers delineate maximum and minimum data points that are no more than 1.5 times the interquartile range, and dots represent individual data points. The notches are defined as ±1.58 times the interquartile range/square root of the sample number, and represent the 95% confidence interval for each median. The average is shown by +.

## Acknowledgements

We thank Drs Yoshinori Ohsumi, Hayashi Yamamoto, and Hitoshi Nakatogawa for discussion and yeast strains, Drs Shuh-ichi Nishikawa and Masahiko Watanabe for antibodies, Drs Yasuyoshi Sakai, Masahide Oku, and Akio Kihara for discussion, and Mr Yasutomo Ito for molecular modeling. This study was supported by Grants-in-Aid for Scientific Research from the Ministry of Education, Culture,

Sports, Science, and Technology of the Government of Japan to Takuma Tsuji (15K18954, 17K15544) and Toyoshi Fujimoto (25111510, 15H02500, 15H05902).

## Additional information

### Funding

| Funder | Grant reference number | Author |
|---|---|---|
| Japan Society for the Promotion of Science | 25111510 | Toyoshi Fujimoto |
| Japan Society for the Promotion of Science | 15H02500 | Toyoshi Fujimoto |
| Japan Society for the Promotion of Science | 15H05902 | Toyoshi Fujimoto |
| Japan Society for the Promotion of Science | 15K18954 | Takuma Tsuji |
| Japan Society for the Promotion of Science | 17K15544 | Takuma Tsuji |

The funders had no role in study design, data collection and interpretation, or the decision to submit the work for publication.

### Author contributions

TTs, Conceptualization, Resources, Supervision, Funding acquisition, Writing—original draft, Project administration, Writing—review and editing; MF, Data curation, Funding acquisition, Investigation, Visualization, Methodology, Writing—review and editing; TTa, Investigation, substantial contribution to the acquisition of data; JC, Resources, Investigation, Visualization, substantial contribution to the acquisition of data; MO, Data curation, Investigation, Visualization; ST, Investigation, substantial contribution to the interpretation of data; TF, Conceptualization, Investigation

### Author ORCIDs

Toyoshi Fujimoto, http://orcid.org/0000-0002-3601-7977

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
