## [Decision Letter]

Thank you for submitting your article "Niemann-Pick type C proteins promote microautophagy by expanding raft-like membrane domains in the yeast vacuole" for consideration by *eLife*. Your article has been favorably evaluated by Randy Schekman (Senior Editor) and three reviewers, one of whom, Suzanne Pfeffer, is a member of our Board of Reviewing Editors.

The reviewers have discussed the reviews with one another and the Reviewing Editor has drafted this decision to help you prepare a revised submission

Summary:

Through a series of elegant and technically challenging morphological assays, the authors present compelling freeze-fracture images showing the vacuolar phenotypes in yeast mutants defective in the Niemann-pick type C (NPC) proteins. The authors show that the NPC proteins play an important role in the formation and expansion of lipid domains on the vacuole membrane in both stationary phase, and after nitrogen starvation (vacuole domain formation has been previously reported: Moeller et at., 1979, Toulmay et al., 2013, and Wang et al., 2014). They use freeze fracture EM and replica labeling to show that LDs enter the vacuole using a microautophagic mechanism which occurs at the sterol-rich, IMP-deficient vacuolar domains. In npc mutants microautophagic lipophagy is strongly inhibited. The authors do observe microautophagic vesicles in the vacuole lumen in npc mutants, however, the vesicles are significantly smaller than those in WT cells, indicating that NPC proteins are required for the expansion of vacuolar domains needed to engulf the relatively large LDs. The authors also address whether macroautophagy and the MVB pathway, both previously shown to be required for vacuolar domain formation and lipophagy, play direct or indirect roles. There are several major points which need to be addressed to support their conclusions.

1) NPC induced sterol trafficking defect (Figure 3 and Figure 5) The authors show a moderate defect in domain formation by ncr1 mutants, and a strong defect in npc2 mutants. The authors attribute this to the lack of cholesterol transport from the vacuole lumen into the vacuole membrane in cells at stationary phase and under nitrogen starvation. The assays used by the authors, however, do not adequately show that the vacuole membrane has lowered sterol concentrations in npc mutants. They instead point to an accumulation of sterols in the vacuole lumen. This is especially apparent in Figure 5, where the WT vacuole membrane does not show strong filipin staining after nitrogen starvation. In addition, the lack of a vacuole membrane marker makes it hard to interpret these data. The authors use lipophagy as a reporter for vacuole domain expansion. However, they do not address whether npc mutants affect the number or size of lipid droplets at stationary phase or under nitrogen starvation. Please provide data relevant to this issue.

2) (Figure 4) The core ATG components have been shown to be required for vacuole domain formation as well as lipophagy, however, their exact role is not clear (Wang et al., 2014). In this manuscript, the authors attribute this defect to two different mechanisms. Under saturation conditions, core atg mutants cause a NPC trafficking defect, re-localizing NPCs from the vacuole to a dot near the vacuole. They claim that this prevents sterol transport to the vacuole membrane, inhibiting domain formation. The authors do not show that vacuole membrane sterol concentration is affected in these cells. Under nitrogen starvation, they do not see a NPC trafficking defect, which is inconsistent with their conclusions about the role of ATGs in saturated cells. They instead posit that autophagy is required for the delivery of sphingolipids to the vacuole membrane. The authors do not consider an alternative mechanism in which macroautophagy is required under starvation conditions for nutrient availability, indirectly blocking the function of other components required for vacuolar domain formation. Please clarify and please explain why Ncr1p and Npc2p are mis-localized when autophagy is disrupted.

3) (Figure 6) The requirement of the MVB pathway for vacuole domain formation, and lipophagy has been previously reported (Toulmay et al., 2013, and Wang et al., 2014). The authors claim that the MVB pathway is required for the delivery of sterols to the vacuole for domain formation. There are three major problems with this conclusion. First, it is not well established in yeast that ILVs are sterol rich. Second, the vps4 mutant used to block ILV formation causes defects in NPC trafficking. Third, the authors do not address whether vacuole membrane sterol distribution is altered in MVB mutants. Please address.

Other comments:

Figure 1—figure supplement 1 should be a regular figure to help a non-expert reader.

Figure 1: "Vph1p-mRFP were found in the IMP-rich region" (please re-state).

Figure 2: "labeling of PI3P WITH GOLD LABELED PX DOMAIN"…

Figure 3: Please label axes on graphs.

Figure 2—figure supplement 3: Please label the structure as "Vacuole" for clarity.

[Editors' note: further revisions were requested prior to acceptance, as described below.]

Thank you for resubmitting your work entitled "Niemann-Pick type C proteins promote microautophagy by expanding raft-like membrane domains in the yeast vacuole" for further consideration at *eLife*. Your revised article has been favorably evaluated by Randy Schekman (Senior Editor) and two reviewers, one of whom, Suzanne Pfeffer, is a member of our Board of Reviewing Editors.

The manuscript has been improved but there are some remaining textual changes that need to be incorporated before acceptance, as outlined below:

1) The main comment about the original version of the manuscript was that the authors did not quantify cholesterol levels at the vacuole membrane, and did not show cholesterol distribution in many of the mutants which they claim block vacuole membrane sterol delivery. In response, the authors quantified the distribution of filipin staining as the percent of cells showing PM and vacuole membrane ("double-ring") staining under saturation conditions. The authors also showed filipin staining for other mutants which they claim "lower vacuole membrane sterol delivery" or affect NPC localization. The data do not actually show lowering of cholesterol, however.

The filipin only shows the relative distribution of sterols in these cells, and does not indicate absolute amounts (increased Vacuolar Membrane sterols under saturation conditions). Another drawback to filipin staining is that it also stains lipid droplets, which under the tested conditions should dock on the vacuole membrane. In many of the images shown, the vacuole membrane filipin staining looks patchy, it is not clear if these dots represent raft-like domains, or perivacuolar sterol rich compartments. We recommend that the authors clarify this explicitly throughout the text and Abstract – they cannot conclude that "loss of NPC lowers the cholesterol concentration of vacuolar membranes."

2) There are way too many supplemental figures. Please combine as many as possible. Figure 3—figure supplement 1 – please state that these are wild type cells in the legend. Figure 5—figure supplement 1 is the same as Figure 5 – please delete. Abstract – please add the words, in stationary phase. Please add mag. bar to 4A.

---

## [Author Response]

1) NPC induced sterol trafficking defect (Figure 3 and Figure 5) The authors show a moderate defect in domain formation by ncr1 mutants, and a strong defect in npc2 mutants. The authors attribute this to the lack of cholesterol transport from the vacuole lumen into the vacuole membrane in cells at stationary phase and under nitrogen starvation. The assays used by the authors, however, do not adequately show that the vacuole membrane has lowered sterol concentrations in npc mutants. They instead point to an accumulation of sterols in the vacuole lumen. This is especially apparent in Figure 5, where the WT vacuole membrane does not show strong filipin staining after nitrogen starvation. In addition, the lack of a vacuole membrane marker makes it hard to interpret these data.

As the reviewers pointed out, in the original manuscript, filipin staining was used to show sterol-rich deposits in the vacuolar lumen of NPC mutants. To examine changes in sterol in the vacuolar membrane, quantification of sterol in isolated vacuole fractions was considered as a possibility, but we found that isolated vacuoles retain macro- and micro-autophagic vesicles as well as ILVs in the lumen, making it difficult to measure the sterol content in the vacuolar membrane per se. We thus utilized filipin staining to examine whether the sterol concentration in the vacuolar membrane of mutant yeast was higher/lower than that of wild-type yeast.

With regard to stationary phase yeast, which generally contains one large vacuole, filipin staining revealed two concentric fluorescent rings, i.e., one in the plasma membrane and one in the vacuolar membrane, in a majority of wild-type cells. This was drastically different from the result in log phase wild-type yeast, in which the vacuolar membrane contains little sterol and filipin staining invariably gives a single-ring fluorescence pattern (Figure 3—figure supplement 1), indicating that the sterol concentration of the vacuole membrane increased in stationary phase. Based on these results, we compared the proportion of cells showing the double-ring filipin staining pattern. The result showed that cells with the double-ring pattern were significantly less abundant in NPC mutants than in wild-type cells, indicating that sterol in the vacuolar membrane is actually lower in NPC mutants. The result of this experiment was added to the text (subsection “Sterol transport by NPC proteins is essential for stationary phase lipophagy”, fifth paragraph) and as the new Figure 3.

For yeast under nitrogen starvation, the vacuolar membrane needs to be visualized with a fluorescent marker because the vacuolar shape is grossly changed, especially in the wild-type yeast, and is therefore not distinct using the Nomarski method. We thus modified the protocol to enable simultaneous observation of the filipin staining and Vph1-GFP. The result showed that the filipin staining in NPC mutants is seen as large puncta within the Vph1p-GFP ring (i.e., in the vacuolar lumen), whereas that in wild-type cells is observed as indiscrete signals largely overlapping with the deformed vacuolar membrane visualized by Vph1p-GFP. The filipin staining in the wild-type vacuolar membrane was less distinct than that in stationary phase. This result agrees well with the freeze-fracture EM result showing that the raft-like domain formation is less extensive in nitrogen starvation than in stationary phase, indicating that the increase in sterol in the vacuolar membrane in nitrogen starvation is not so drastic as in stationary phase. The method of this experiment and its result were added to the text (subsection “Fluorescence microscopy”, second paragraph and subsection “Microautophagy in acute nitrogen starvation also occurs in NPC-dependent raft-like domains”, second paragraph, respectively) and as the new Figure 5.

*The authors use lipophagy as a reporter for vacuole domain expansion. However, they do not address whether npc mutants affect the number or size of lipid droplets at stationary phase or under nitrogen starvation. Please provide data relevant to this issue.*

The number and size of lipid droplets in NPC mutants were examined by fluorescence microscopy after BODIPY493/503 staining and by freeze-fracture EM, respectively. The result showed that both the number and size of lipid droplets are not drastically different between wild-type cells and NPC mutants, both in stationary phase and under nitrogen starvation. The result was added to the text (subsection “Sterol transport by NPC proteins is essential for stationary phase lipophagy”, third paragraph; subsection “Microautophagy in acute nitrogen starvation also occurs in NPC-dependent raft-like domains”, fourth paragraph) and also as Figure 3—figure supplement 2 and Figure 5—figure supplement 4.

*2) (Figure 4) The core ATG components have been shown to be required for vacuole domain formation as well as lipophagy, however, their exact role is not clear (Wang et al., 2014). In this manuscript, the authors attribute this defect to two different mechanisms. Under saturation conditions, core atg mutants cause a NPC trafficking defect, re-localizing NPCs from the vacuole to a dot near the vacuole. They claim that this prevents sterol transport to the vacuole membrane, inhibiting domain formation. The authors do not show that vacuole membrane sterol concentration is affected in these cells.*

The sterol concentration in the stationary phase vacuole in *atg* mutants was examined by quantifying the double-ring filipin staining pattern as described above in our first answer to comment 1. The result indicated that the sterol concentration in the vacuolar membrane of *atg7Δ* is significantly decreased compared to that in wild-type cells. This result was described in the text (subsection “Trafficking defect of NPC proteins impairs stationary phase lipophagy”, second paragraph) and added as Figure 4—figure supplement 3.

*Under nitrogen starvation, they do not see a NPC trafficking defect, which is inconsistent with their conclusions about the role of ATGs in saturated cells. They instead posit that autophagy is required for the delivery of sphingolipids to the vacuole membrane. The authors do not consider an alternative mechanism in which macroautophagy is required under starvation conditions for nutrient availability, indirectly blocking the function of other components required for vacuolar domain formation. Please clarify.*

As pointed out correctly by the reviewers, a possible deficiency of sphingolipids due to the absence of autophagosomal fusion is one of the possible causes contributing to the observed decrease of vacuolar domain formation in *atg* mutants. In the revised manuscript, we discussed other possible abnormalities leading to the decrease in vacuolar domain formation: transport of sphingolipids to the vacuole by pathways other than autophagosomes may be down-regulated in *atg* mutants; and the vacuole-associated actin network may not be properly organized in *atg* mutants. These possibilities were discussed in the revised manuscript (subsection “Correlation of macroautophagy and microautophagy in nitrogen starvation”, last two paragraphs).

*Please explain why Ncr1p and Npc2p are mis-localized when autophagy is disrupted.*

We think that the aberrant distribution of NPC proteins in stationary phase *atg* mutants may not be directly derived from the lack of macroautophagy, because the distribution persisted in *atg18Δ* even when the autophagic activity was restored by expression of Atg18(FTTG)-2xFYVE, which binds to PtdIns(3)P, but not to PtdIns(3,5)P_2_ (Figure 4). Furthermore, *fab1Δ*, which lacks PtdIns(3,5)P_2_ but retains macroautophagic activity, also showed similar NPC puncta in stationary phase (Figure 4). Based on these results we conjectured that some abnormality in the PtdIns(3,5)P_2_ metabolism may occur in *atg* mutants, thereby leading to the abnormal NPC distribution.

The above conjecture was supported by the additional experiments. By studying the time course of the puncta formation in *atg7Δ* and in other mutants, it was found that the abnormal NPC puncta in *atg7Δ* were not observed in day 1, a period corresponding to the early post-diauxic phase, appeared in some cells in day 2, and finally became prevalent in day 3 when cells reached the stationary phase. In contrast, in *pep4Δ*, which is defective in recycling of autophagocytosed materials, NPC proteins showed the vacuolar distribution even in stationary phase, arguing that nutritional deficiency caused by the absence of macroautophagy may not be the direct cause of their aberrant distribution. Furthermore, in *fab1Δ*, which is totally lacking in PtdIns(3,5)P_2_, NPC proteins started to show the punctate distribution in a majority of cells in day 1. These results were added as Figure 4—figure supplement 1 and discussed in the text (subsection “Trafficking defect of NPC proteins impairs stationary phase lipophagy”) in the revised manuscript.

These results suggested that abnormality in the phosphoinositide metabolism, in particular that of PtdIns(3,5)P_2_, may occur in *atg* mutants, causing the aberrant distribution of NPC proteins. But we are still seeking the mechanism causing the defective PtdIns(3,5)P_2_ metabolism, as well as how such abnormality leads to the trafficking defect of NPC proteins. We think that the peculiar behavior of NPC proteins in *atg* mutants is worth further investigation and wish to clarify the details in future studies.

*3) (Figure 6) The requirement of the MVB pathway for vacuole domain formation, and lipophagy has been previously reported (Toulmay et al., 2013, and Wang et al., 2014). The authors claim that the MVB pathway is required for the delivery of sterols to the vacuole for domain formation. There are three major problems with this conclusion. First, it is not well established in yeast that ILVs are sterol rich.*

We examined *atg7Δpep4Δprb1Δ* in nitrogen starvation. In this mutant, the vacuolar lumen was expected to contain abundant ILVs due to the absence of proteolytic degradation, whereas structures other than ILVs must be scarce because both macroautophagy and microautophagy are abrogated. This assumption was confirmed by freeze-fracture EM, which showed the presence of numerous ILVs with few other structures (added as Figure 6—figure supplement 4). In contrast, the vacuolar lumen of *atg7Δ* was largely vacant (the former Figure 7—figure supplement 2 was moved to Figure 6—figure supplement 4). When *atg7Δpep4Δprb1Δ* and *atg7Δ* under nitrogen starvation were compared by filipin staining, the vacuolar lumen of *atg7Δpep4Δprb1Δ* was filled with filipin fluorescence, whereas that of *atg7Δ* was not. Although individual ILVs cannot be distinguished by fluorescence microscopy, the combined result indicates that yeast ILVs are enriched with sterol. This result was described in the text (subsection “The MVB pathway supplies sterol for induction of microautophagy”, second paragraph) and the filipin staining result was added as Figure 6.

*Second, the vps4 mutant used to block ILV formation causes defects in NPC trafficking.*

We confirmed that Ncr1p-GFP and Npc2p-GFP in *vps4Δ* after nitrogen starvation were observed in the membrane and the lumen of the vacuole, respectively, as they were in the wild-type cell. The result in *vps4Δ* contrasted with the aberrant distribution of both Ncr1p-GFP and Npc2p-GFP in *vps4Δ* in stationary phase (Figure 4—figure supplement 7). The result in *vps4Δ* after nitrogen starvation was added to the text (subsection “The MVB pathway supplies sterol for induction of microautophagy”, second paragraph) and as Figure 6—figure supplement 3 in the revised manuscript.

*Third, the authors do not address whether vacuole membrane sterol distribution is altered in MVB mutants. Please address.*

By using the protocol described in the first answer to comment 1 [the last paragraph of that answer], filipin staining in *vps4Δ* was examined. In *vps4Δ* under nitrogen starvation, Vph1-GFP was observed as a large dot representing the class E compartment as well as in the vacuolar membrane, and the vacuole largely retained its round shape. In comparison to wild-type cells, which showed filipin staining overlapping with the Vph1-GFP-positive vacuolar membrane, filipin-derived fluorescence in *vps4Δ* was intense in the class E compartment but negligible in the vacuolar membrane. The result indicated that the sterol concentration in the vacuolar membrane may not increase in *vps4Δ* in nitrogen starvation. This result was added to the text (subsection “The MVB pathway supplies sterol for induction of microautophagy”, second paragraph) and as Figure 6—figure supplement 2.

*Other comments:*

Figure 1—figure supplement 1 should be a regular figure to help a non-expert reader.

The former Figure 1—figure supplement 1 was moved into the main figure as Figure 1 in the revised manuscript.

Figure 1: "Vph1p-mRFP were found in the IMP-rich region" (please re-state).

According to the suggestion, the legend for the figure (Figure 1 in the revised manuscript) now reads as follows:

“Labels for Vph1p-mRFP (arrows) were excluded from the polygonal IMP-deficient area, but were present in the IMP-rich region, indicating that the IMP-deficient area corresponds to the raft-like domain.”

*Figure 2: "labeling of PI3P WITH GOLD LABELED PX DOMAIN"…*

According to the suggestion, the legend for Figure 2 now reads as follows:

“Freeze-fracture replica labeling of PtdIns(3)P in stationary phase vacuole. Colloidal gold particles indicate PtdIns(3)P labeled by recombinant GST-p40^phox^ PX domain.”

*Figure 3: Please label axes on graphs.*

Axes in Figure 3 were labeled as “Proportion of cells in respective categories (%)”.

*Figure 2—figure supplement 3: Please label the structure as "Vacuole" for clarity.*

The vacuole in the diagram is now labeled as such in Figure 2—figure supplement 3.

[Editors' note: further revisions were requested prior to acceptance, as described below.]

*The manuscript has been improved but there are some remaining textual changes that need to be incorporated before acceptance, as outlined below:*

*1) The main comment about the original version of the manuscript was that the authors did not quantify cholesterol levels at the vacuole membrane, and did not show cholesterol distribution in many of the mutants which they claim block vacuole membrane sterol delivery. In response, the authors quantified the distribution of filipin staining as the percent of cells showing PM and vacuole membrane ("double-ring") staining under saturation conditions. The authors also showed filipin staining for other mutants which they claim "lower vacuole membrane sterol delivery" or affect NPC localization. The data do not actually show lowering of cholesterol, however.*

*The filipin only shows the relative distribution of sterols in these cells, and does not indicate absolute amounts (increased Vacuolar Membrane sterols under saturation conditions). Another drawback to filipin staining is that it also stains lipid droplets, which under the tested conditions should dock on the vacuole membrane. In many of the images shown, the vacuole membrane filipin staining looks patchy, it is not clear if these dots represent raft-like domains, or perivacuolar sterol rich compartments. We recommend that the authors clarify this explicitly throughout the text and Abstract – they cannot conclude that "loss of NPC lowers the cholesterol concentration of vacuolar membranes."*

We agree with the reviewers’ comment in that filipin staining shows only the relative amount of sterol in cellular membranes, and that some of the staining in the vacuolar region might be derived from other membranes near the vacuole (although sterol esters in lipid droplets are not stained by filipin). In the revised manuscript, we discussed these problems in the filipin staining experiment and toned down the sentences regarding the sterol concentration. We also added an EM photo to show the presence of filipin-sterol complexes in the stationary phase vacuolar membrane (Figure 3—figure supplement 1). [please see the following 2) for the context to which this figure was added]

The changed portions are as follows:

1) Abstract: “The result shows that a prime function of NPC proteins is to promote microautophagy *probably* by increasing sterol in the vacuolar membrane.”

2) Results: “Consistent with the formation of raft-like domains, sterol stained by filipin was observed along the vacuolar membrane in stationary phase, giving a double-ring staining pattern, whereas filipin staining in log-phase yeast was found only in the cell surface (Figure 3—figure supplement 1). […] We hypothesized that this *probable* increase in sterol in the stationary phase vacuolar membrane may not occur through spontaneous incorporation but may involve NPC proteins”.

3) Results: “Fourth, the proportion of cells showing the double-ring filipin staining pattern was significantly lower in NPC-deficient cells than in WT cells, indicating that sterol transport to the vacuolar membrane may be suppressed in NPC-deficient cells (Figure 3).”

4) Results: “Consistent with the decrease in NPC proteins in the vacuole, the proportion of cells showing the double-ring filipin staining pattern was lower in *atg7△* than in WT (Figure 4—figure supplement 3).”

5) Results: “Filipin staining suggested a relative increase in sterol in the vacuolar membrane of WT”.

6) Discussion: “The present study showed that NPC proteins play a crucial role in expansion of raft-like vacuolar membrane domains in microautophagy, probably by transporting sterol to the vacuolar membrane.”

*2) There are way too many supplemental figures. Please combine as many as possible.*

Supplemental figures were combined as much as possible. They were reduced from 32 figures to 19 figures. We hope this number of supplemental figures is acceptable.

*Figure 3—figure supplement 1 – please state that these are wild type cells in the legend.*

The legend was revised to state that the cells shown in Figure 3—figure supplement 1 are wild-type yeast.

*Figure 5—figure supplement 1 is the same as Figure 5 – please delete.*

Figure 5—figure supplement 1 was deleted.

*Abstract – please add the words, in stationary phase.*

We understand that this comment is on the role played by the multivesicular pathway in microautophagy during acute nitrogen starvation. We revised the Abstract accordingly.

*Please add mag. bar to 4A.*

A scale bar was added to the left panel of Figure 4.